# Research

 

**Cite this article:** de Framond L, Brumm H, Thompson WI, Drabing SM, Francis CD. 2022 The broken-wing display across birds and the conditions for its evolution. *Proc. R. Soc. B* **289**: 20220058.

behaviour, ecology, evolution

anti-predator behaviour, deception, distraction, nest defence, nest predation, phylogenetic

**Author for correspondence:**
Clinton D. Francis
e-mail: cdfranci@calpoly.edu

†Present address: Max Planck Institute for Biological Intelligence, in foundation, Seewiesen, Germany.

# The broken-wing display across birds and the conditions for its evolution

Léna de Framond[1,†], Henrik Brumm[1,†], Wren I. Thompson[2], Shane M. Drabing[2] and Clinton D. Francis[1,2]

[1]Communication and Social Behavior Group, Max Planck Institute for Ornithology, Seewiesen, Germany
[2]Department of Biological Sciences, California Polytechnic State University, San Luis Obispo, CA, USA

HB, 0000-0002-6915-9357; WIT, 0000-0001-8246-4404; SMD, 0000-0001-9521-6610; CDF, 0000-0003-2018-4954

The broken-wing display is a well-known and conspicuous deceptive signal used to protect birds' broods against diurnal terrestrial predators. Although commonly associated with shorebirds, it remains unknown how common the behaviour is across birds and what forces are associated with the evolution of the display. Here, we use the broken-wing display as a paradigmatic example to study the evolution of a behaviour across Aves. We show that the display is widespread: it has been described in 52 families spread throughout the phylogeny, suggesting that it independently evolved multiple times. Further, we evaluated the association with 16 ecological and life-history variables hypothesized to be related to the evolution of the broken-wing display. Eight variables were associated with the display. We found that species breeding farther from the equator, in more dense environments, with shorter incubation periods, and relatively little nest cover were more likely to perform the display, as were those in which only one parent incubates eggs, species that mob nest predators and species that are altricial or multi-brooded. Collectively, our comprehensive approach identified forces associated with the repeated evolution of this conspicuous display, thereby providing new insights into how deceptive behaviours evolve in the context of predator–prey interactions.

## 1. Introduction

Selection usually reinforces honest animal signals [1], but those that are not honest, such as deceptive signals and displays, are among the most renowned in nature. Many of the best-known deceptive displays appear in response to attempted predation. One famous example is seen in young Virginia opossums (*Didelphus virginiana*), which assume a prone and paralysed position (i.e. playing dead) to avoid predation and increase chances of escape [2]. Burrowing owls (*Athene cunicularia*) employ deception in a different manner; they protect broods from ground squirrels by producing a rattlesnake-mimicking hiss [3]. The broken-wing display (figure 1) is another well-known example. Rather than using crypsis or aggression to defend eggs or a brood, bird parents conspicuously display a feigned injury to lure predators away from vulnerable offspring or eggs [4].

Naturalists have long noted this eye-catching behaviour and it appears in the literature at least as early as 1861 [5] and possibly almost 100 years earlier in Buffon's [6] description of a feigned injury in pied avocets (*Recurvirostra avosetta*). Because the behaviour is composed of suggestive movements and the injury-feigning bird flies from the ground when the predator is lured away, the broken-wing display is widely accepted to be directed towards diurnal, terrestrial predators [7] and often associated with shorebirds [8]. Indeed, Gomez-Serrano [9] states that the display reaches its 'pinnacle' in shorebirds and defines the broken-wing display as 'a feigning behaviour that some

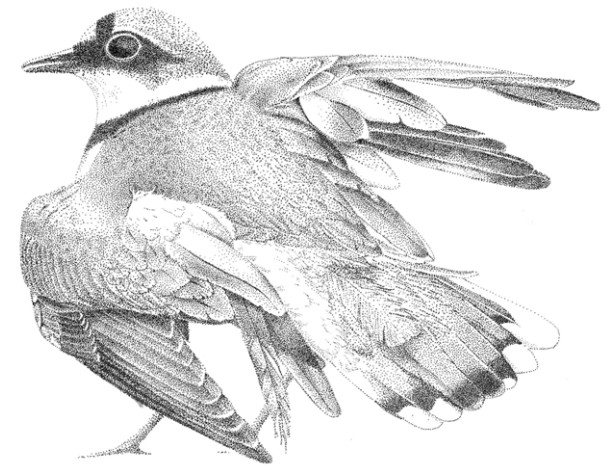

**Figure 1.** The broken-wing display. Depicted is a common ringed plover (*Charadrius hiaticula*). Illustration by L. de Framond.

ground-nesting birds perform to lure a potential predator away from the bird's nest or chicks'. However, this deceptive display has also been observed for other species across the class Aves (reviewed in [4]) with traits and nesting habits that are very different from shorebirds, raising important questions about the phylogenetic distribution of the display and the selective agents responsible for its evolution. Is the behaviour predominantly found in shorebirds, as colloquial knowledge would suggest, or is it more widespread? Are there particular ecological contexts or life histories that are associated with the display?

Here, we investigated the broken-wing display as an example for the study of evolution of a behavioural trait across an entire class of vertebrates. Injury-feigning displays have also been reported across terrestrial vertebrates [10], but only among birds do such a display appears to be widespread [4]. If the broken-wing display convergently evolved in unrelated avian taxa, a comparative study may reveal factors associated with the evolution of the behaviour and thus improve our understanding of behavioural adaptation. We first provided a comprehensive assessment of the phylogenetic distribution of the broken-wing display across class Aves. Then, using a dataset of species documented to perform the display balanced with closely related species that appear to not perform the display, we determined the ecological and life-history contexts and traits that explain variation in the broken-wing display using phylogenetic logistic regression. In these analyses, we test our own hypotheses regarding the forces that might select for, or coevolve with, the use of the broken-wing display, plus several others proposed over the last seven decades. We organized our predictor variables into four, non-mutually exclusive categories: life history, predation risk, investment in current reproduction and future reproductive potential (table 1).

## 2. Methods

### (a) Data collection

To build a database of species that perform the broken-wing display, we conducted literature searches in 2016 and 2018 using Google Scholar and Web of Science with different combinations of the search terms 'broken wing', 'display',

'behaviour', 'feigned injury' and 'distraction display'. We also searched the Handbook of Birds of the World (HBW) [12,13] with the same terms. We then evaluated records for descriptions of the behaviour. We considered species to perform the display when descriptions clearly described distraction displays involving the dragging of wings or other feigned wing injuries and excluded other distraction displays (e.g. rodent run, diversionary flight, etc.) [4].

Because the literature may be incomplete regarding which species perform the broken-wing display, we distributed a survey to professional ornithologists, avian ecologists and experienced birders in 2016. The survey prompted participants to specify the species and environmental context of the observation of the display and was distributed to major international ornithological centres/museums, posted to ornithological society email lists and advertised at an international ornithological conference (electronic supplementary material). The majority of species performing the display were documented in the literature (225 species). The survey added another 61 species, 37 of which were subsequently confirmed to perform the broken-wing display with targeted literature searches (electronic supplementary material).

### (i) Database for testing traits related to occurrence of broken-wing display

To evaluate the ecological or life-history traits associated with the occurrence of the broken-wing display, for each of the 285 species documented to perform the broken-wing display, we matched it to a close relative with no documentation of exhibiting the display to achieve balanced representation across clades, which is an approach derived from Felsenstein's phylogenetic contrasts approach [14] and widely used in comparative studies (e.g. [15,16]). Specifically, using a consensus phylogeny based on Jetz *et al.* [17], for each species known to perform the display, we found the closest relative that does not perform the display and was not already included in the database as a close relative to another species documented to perform the display.

*Life-history traits.* We obtained body mass from Dunning *et al.* [18] and used the natural logarithm of mass as a predictor in models. We obtained the minimum and maximum latitude of each species' breeding range using range shape files from BirdLife International [19]. We included the resident range for non-migratory species, the breeding range for migratory species and the breeding range and resident range for species with populations that are migratory and non-migratory. For species where the range was not available in this database, we manually measured the maximum and minimum latitude of their range based on the range maps provided in the IUCN Red List (https://www.iucnredlist.org/). As a proxy for agents of selection that vary with latitude, we used the absolute value of either the minimum or maximum latitude, whichever was larger. This measure has the advantage that it cannot result in an uninhabited location, which can occur with species with discontinuous ranges when a midpoint is used [20].

To assess species' colonial habits, we screened the 'breeding' section in HBW [12] and the 'Birds of the World' (BOW; https://birdsoftheworld.org/bow/home), and, when available, the 'social and interspecific interaction' part of the 'behaviour' section in HBW. Species' colonial habits were scored on a scale from 0 (solitary) to 2 (colonial). Because colonially breeding species are very conspicuous, we assumed that species for which no description was available are solitary. Clearly colonial species were assigned a score of 2, semi-colonial, sometimes colonial, and loose groups/colony species were assigned a score of 1. Species with mentions of rare, loose groups or with a complex socio-reproductive system (e.g. polygyny, polyandry, pairs with helper males) were assigned a score of 0. We scored precociality using assignments of altricial or precocial in HBW.

**Table 1.** Summary of the hypotheses, predictions and whether they have been proposed in the literature. Predictors are organized by four non-mutually exclusive categories. Predictions + and − denote positive or negative associations of the trait with evolution of the broken-wing display.

| category | predictor | prediction | rationale | reference |
|---|---|---|---|---|
| life history | precocial chicks | + | more precocial species, such as many shorebirds, are well-known to perform the broken-wing display, thus this life-history strategy may correlate with use of the display | |
| | body mass | − | larger species are more effective in defending nests from predators through aggression, thus smaller species may use other strategies in nest defence | [11] |
| | colonial species | − | nest density is too high to deceive a predator from finding nests | [7] |
| predation risk | nest cover | − | concealed nests that are less visible are less likely to be discovered by visual predators | [7] |
| | nest on or near ground | + | nests close to the ground are more accessible to most predators | [7] |
| | nest protection | − | nests that provide physical protection for eggs and chicks should be less susceptible to predation | |
| | incubation duration | − | if the broken-wing display evolves in response to intensity of predation pressure, the display should negatively covary with duration of the incubation period, which tends to be shorter for species that experience high predation rates | |
| | absolute latitude | + | high absolute latitudes are associated with longer daylight, which will benefit terrestrial diurnal predators through extended search time | [7] |
| | habitat density | − | brooding parents can detect predators early in more open environments and move away from the nest to use distraction displays | [7] |
| | nest conspicuousness | − | distraction displays may be ineffective for highly conspicuous nests that visual predators can easily detect | [7] |
| investment in current reproductive attempt | number of eggs | + | higher energetic investment in a single nesting attempt should select for strategies to maximize survival of current attempt | |
| | mass of clutch relative to bird body mass | + | | |
| | incubation duty | − | because of high risks, two birds are more likely to exhibit aggressive nest defence compared to a single bird. Thus, a single bird may use the broken-wing display as a less risky tactic | [11] |
| | mobbing | − | risk trade-offs between mobbing and the broken-wing display may favour one or the other | [11] |
| future reproductive potential | longevity | − | birds with high future reproductive potential may prioritize survival over any form of nest defence | |
| | double brooding | − | | |

*Traits related to predation risk.* We obtained details on nest location, type and cover from HBW. Nest location was placed into four categories: ground, near ground (less than or equal to 3 m), above ground and on or above water. Based on descriptions of nest placement, we created a nest cover index from 1 to 3: 1 denoted little to no vegetation that would visually conceal the nest, 2 denoted partial visual concealment by vegetation and 3 reflected placement where dense vegetation would nearly or completely conceal the nest. Nest type was assigned to one of six categories: cavity, cup, dome, mound, platform, scrape and stick. Because we hypothesized that nest types provide different levels of physical protection against predators to eggs and juveniles, irrespective of placement in relation to the ground, we assigned each nest type to an index score spanning 0–2. As

scrape nests do not provide any physical barrier between the nest content and potential predators, they were assumed to provide no protection against predation and assigned the value 0. Stick, cup and platform nests were assumed to provide an intermediate level of protection as the nest contents are partially enclosed and assigned the value 1. Finally, cavity, mound and dome nests were assigned the value 2 because they are fully enclosed and may provide the most protection against predation. To reflect our alternative hypothesis that the physical structure of the nest could attract visual predators, we also assigned nests index values ranging 0–5 based on conspicuousness of the nest's physical structure: scrape and cavity nests were least conspicuous (no apparent nest structure) and assigned a value of 0, mound nests were most conspicuous (very large nest structure) and assigned a value of 5, and stick, dome, cup and platform nests were of gradually increasing conspicuousness.

Species habitat affiliations can be difficult to summarize given the range of habitat conditions that some species use. To overcome this challenge, we created an index that incorporated the nuance of habitat association by converting HBW habitat descriptions into a habitat score. We categorized the words describing characteristic features of habitats in the descriptions into seven classes of increasing vegetation densities and assigned each category a score. The occurrence of words belonging to each category was used to get an average vegetation density score. Based on a subset of species, we verified that this average vegetation density score was strongly correlated with average per cent forest cover from a recently published database [21] (electronic supplementary material).

*Traits related to current reproduction investment.* We measured the investment in current reproduction using the number and relative mass of eggs, incubation characteristics and the defence of the brood. Mean clutch size was collected from the 'breeding' or 'demography' sections of BOW. When only clutch size range was reported, we calculated the midpoint. Egg length, breadth and mass were collected simultaneously. In 207 species, only length and breadth were available. Following Hoyt [22], we calculated mass of fresh eggs as follows:

$$\text{mass} = \text{Kw} \times \text{length} \times \text{breadth}^2. \tag{2.1}$$

Because constant Kw is species-specific, but shows only minor variation among species, we used the mean of all values of Kw provided in Hoyt [22] (i.e. Kw = 0.548). To test this approach, we compared fresh egg mass calculated with this method to values for fresh mass measurements from BOW for 133 species and found the approach to be a good approximation of egg mass ($r = 0.975$, $p < 0.001$). Total clutch mass was calculated by multiplying egg mass values or estimates by clutch size. We then obtained the relative mass of clutch indexed to body mass of the bird to account for size variation among species.

We recorded the number of parents involved in incubation from HBW and scored as either 1 or 2. The presence of aggressive nest defence behaviour was collected from the 'breeding' and 'behaviour' sections of BOW. Because birds that aggressively defend nests are conspicuous, we assumed that no mention of this behaviour reflects the absence of aggressive nest defence. All mentions of aggressive defence of broods or nests (e.g. 'mobs predator' and 'aggressive scolding') were counted as a presence of aggressive defence, except when only alarm calls are emitted.

*Traits related to future reproduction potential.* Future reproduction depends on the birds' longevity and ability to raise additional broods within a single breeding season. We collected the maximum longevity recorded in marked wild birds from BOW and noted whether species were single or multi-brooded. Species that lay replacement clutches but only raise a single brood to independence each season were considered single-brooded. Species for which reproduction depends on the occurrence of rainfall were recorded as single-brooded, because these species usually raise only one brood per rain event.

## (b) Statistical analysis

To determine the effect of predictors on the probability of performing the broken-wing display, we used phylogenetic generalized logistic models (phylolm, v. 2.6.2) [23] in R (v. 4.0.4, The R Foundation for Statistical Computing). For our phylogenetic hypothesis, we used a consensus tree based on Jetz *et al.* [17], and to improve model convergence, we relaxed the constraints on the phylogenetic model by expanding the parameter search space (btol = 30).

Because trait data were not available for all species, we used a series of models beginning with the most complete predictor variables and then iteratively subset the data to include the predictor with the next highest sample size. Therefore, we first ranked the predictor variables in decreasing order of sample size and at each step included the set of variables with the next highest sample size. Each model was simplified using backwards model selection to include only predictor variables with some evidence of an influence on performance of the broken-wing display. That is, we removed predictors one-at-a-time based on the largest *p*-value and retained predictors where $p \leq 0.1$, which we considered informative predictors. Informative predictors were carried down to the next modelling step for inclusion in the model with the next predictor variables and we again removed predictors one-at-a-time as before (see electronic supplementary material for an example). This sequence was repeated to consider all predictors for which we had data for at least 200 species. When predictors had been informative at larger sample sizes, but not in models with smaller sample sizes, they were removed from the model to avoid unnecessary increasing model complexity. In total, we ran 25 versions of 10 different models (table 2; electronic supplementary material, table S2). We checked collinearity among predictors at each model step and for the final, simplified model at each sample size using the *check_collinearity* function in the performance R package (v. 0.7.0) [24]. We also inspected model residual distributions and checked for outliers, but found none. In the results, we present parameter effect sizes from the model in which it appears with the highest sample size. To visualize results, we calculated confidence intervals for effects by bootstrapping models 100 times. Garland and Ives [25] and Ho *et al.* [23] recommend a high number of bootstrap replicates. However, in sub-analyses, we confirmed that running 100 bootstrap replicates resulted in qualitatively identical confidence intervals for replicates spanning 50–2000 iterations (electronic supplementary material, figure S2).

## 3. Results

We identified 285 species representing 172 genera and 52 families that perform the broken-wing display (figure 2; electronic supplementary material, figures S3–S7). It is exhibited among species spanning the most basal clades of Aves, such as tinamous (Tinamidae), pheasant, quail and allies (Phasianidae), and ducks and geese (Anatidae) to the most derived passerines, such as New World blackbirds (Icteridae), cardinals (Cardinalidae) and New World warblers (Parulidae; figure 2). Our mapping of the phylogenetic distribution of the display reveals clear patterns of clustering within some clades and conspicuous absences within others. For instance, among passerines, the clade within Sylvioidea spanning families Sylviidae, Pycnonodidae, Cisticolidae, Timaliidae and Zosteropidae all have members that perform the broken-wing display (figure 2). Additionally, the display was prominent among shorebirds and other families within Charadriiformes where 10 of 17 families have members that perform the display. Two groups with conspicuous absences are (i) trogons (Trogonidae) and their sister taxon (including

**Table 2.** Results from reduced models for each sample size (see electronic supplementary material, table S2) explaining the probability of performing the broken-wing display. *n* = number of species included in the analysis, 2.5% and 97.5% CI = upper and lower limits of the bootstrap confidence interval.

| model | *n* | predictor | estimate | s.e. | *z* | 2.5% CI | 97.5% CI | *p* |
|-------|-----|-----------|----------|------|-----|---------|----------|-----|
| M0 | 569 | abs. max. latitude | 0.03 | 0.00 | 5.59 | 0.02 | 0.03 | <0.001 |
| | | habitat score | 0.10 | 0.04 | 2.29 | 0.03 | 0.14 | 0.022 |
| | | mobbing | 0.53 | 0.28 | 1.89 | 0.44 | 0.58 | 0.059 |
| M2 | 523 | abs. max. latitude | 0.02 | 0.01 | 4.35 | 0.02 | 0.03 | <0.001 |
| | | habitat score | 0.29 | 0.06 | 5.13 | 0.20 | 0.40 | <0.001 |
| | | nest cover | −0.83 | 0.15 | −5.44 | −1.01 | −0.75 | <0.001 |
| M4 | 439 | habitat score | 0.30 | 0.07 | 4.51 | 0.19 | 0.44 | <0.001 |
| | | nest cover | −1.04 | 0.19 | −5.56 | −1.50 | −0.73 | <0.001 |
| | | precociality | −0.51 | 0.26 | −1.95 | −0.95 | −0.01 | 0.051 |
| M5 | 388 | habitat score | 0.31 | 0.07 | 4.43 | 0.20 | 0.44 | <0.001 |
| | | nest cover | −1.05 | 0.20 | −5.30 | −1.45 | −0.69 | <0.001 |
| | | duty | −0.53 | 0.25 | −2.10 | −1.04 | −0.05 | 0.036 |
| M6 | 340 | habitat score | 0.31 | 0.08 | 3.87 | 0.15 | 0.48 | <0.001 |
| | | nest cover | −0.99 | 0.22 | −4.47 | −1.33 | −0.66 | <0.001 |
| | | incub. duration | −0.05 | 0.02 | −2.15 | −0.06 | −0.03 | 0.032 |
| M8 | 286 | habitat score | 0.34 | 0.08 | 3.99 | 0.22 | 0.50 | <0.001 |
| | | nest cover | −0.95 | 0.22 | −4.33 | −1.38 | −0.49 | <0.001 |
| | | multi-brood | 0.56 | 0.27 | 2.06 | −0.03 | 1.15 | 0.039 |

kingfishers, rollers, woodpeckers, barbets, hornbills, hoopoes) and (ii) turacos (Musophagidae) and the Ardeae clade (e.g. tropicbirds, loons, albatrosses, cormorants and pelicans; figure 2).

## (a) Ecological and life-history correlates of the broken-wing display

Of 16 predictors tested, eight were significantly correlated with the probability to perform the broken-wing display: maximum absolute latitude, habitat score, nest cover, incubation duration, precociality, incubation duty, double brooding and mobbing (figures 3 and 4, and table 2; electronic supplementary material, table S2).

### (i) Life-history traits

Two of three predictors related to life-history traits, body mass and colonial habits were not related to the probability of performing the broken-wing display and dropped from further analysis. Contrary to our prediction, species with precocial chicks were less likely to perform the broken-wing display (60% of precocial species) than species with altricial chicks (67.6% of altricial species, table 2: M4, *n* = 445, figures 3*e* and 4). Neither coloniality (table 2, M0, *n* = 569) nor the natural logarithm of body mass was related to the probability of performing the broken-wing display (table 2: M1, *n* = 537).

### (ii) Traits related to predation risk

Of the seven predation risk predictors, habitat score, incubation duration, nest cover and latitude were all related to the probability of performing the broken-wing display, whereas nest protection, conspicuousness and height were

not (figure 4). Distance from the equator was positively related to the probability of performing the display. For instance, model marginal effect estimates suggest 33.6% of species with maximum range limits up to 30° perform the display, whereas 59.6% of species with maximum range limits between 50 and 80° (table 2: M0, *n* = 569, figure 3*a*). We confirmed this pattern was not an artefact of undersampling in tropical regions by considering only species with maximum range limits of over 30° (polar and temperate regions only) and found that the positive relationship persisted (*n* = 196, *β* = 0.058, *p* < 0.001). Increased incubation duration was associated with a lower probability of performing the broken-wing display (table 2: M6, *n* = 340, figure 3*d*). Specifically, the probability of performing the display declined from greater than 0.8 at incubation durations of less than 20 days to a probability of 0.58 when incubation duration is greater than 30 days. Birds living in more open habitats were less likely to perform the broken-wing display than birds living in more cluttered or forested areas (table 2: M1, *n* = 569, figures 3*b* and 4). However, the effect was not particularly strong: 46.5% of species with habitat scores less than 3 perform the broken-wing display and this only increased to 51.1% for species with habitat scores greater than 5. This positive relationship with habitat density held in a sensitivity analysis restricted to passerine species with data on percentage of forest cover (*n* = 165, *β* = 0.016, *p* = 0.012). Of all nest characteristics considered, only nest cover was related to the probability of performing the broken-wing display. Model marginal effect estimates suggest that the probability of performing the broken-wing display declines from approximately 25% among species that do not conceal their nests to less than 10% for those with completely concealed nests (table 2: M2, *n* = 523, figure 3*c*). Nest

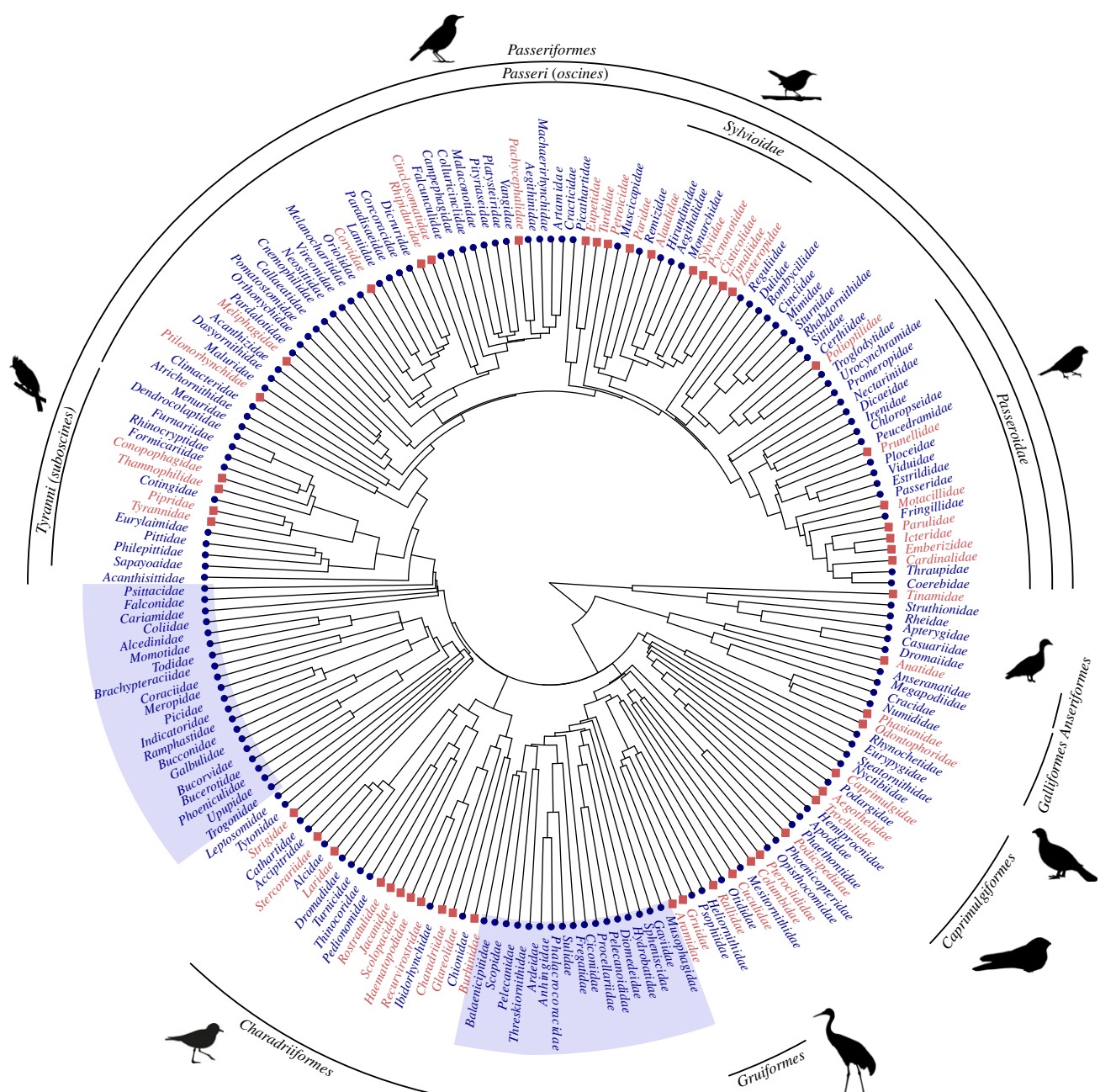

**Figure 2.** Phylogenetic distribution of the performance of the broken-wing display across Aves. Displayed is a simplified family-level tree following the taxonomy in Jetz *et al.* [17] to more easily visualize which families have at least one species that performs the display (red squares) and those where the display has not been documented (blue circles). Select clades labelled to aid interpretation. Those shaded in blue are large, conspicuous clades that do not exhibit the display. (Online version in colour.)

location above ground, nest protection index and nest conspicuousness were uninformative and dropped from the models (M2, $n = 523$, all $p > 0.20$, figure 4).

### (iii) Traits related to the investment in current reproductive effort

Of the four predictors related to reproductive effort, two were associated with the occurrence of the broken-wing display. Supporting our prediction, species in which only one parent incubates were more likely to perform the broken-wing display (65.0%) than species where both parents incubate (59.8%; table 2: M5, $n = 388$, figures 3g and 4). However, in contrast with our hypothesis, species that mob or attack nest predators were more likely to perform the display (68.8%) than non-aggressive species (47.0%, table 2: M0, $n = 569$, figures 3h and 4). This correlation was the only one

that did not hold in a sensitivity analysis in which only species with known clutch size (therefore, low probability of including non-reported displaying species) were used (electronic supplementary material, table S3). Clutch size (M3, $p = 0.92$, $n = 504$) or clutch mass relative to body mass (M7, $p = 0.23$, $n = 260$) were both unrelated to the probability of performing the broken-wing display (figure 4).

### (iv) Traits related to future reproductive potential

Contrary to our predictions, multi-brooded birds were more likely to perform the broken-wing display than single-brooded species (table 2: M8, $\beta = 0.56$, $p = 0.039$, $n = 286$, figures 3f and 4). This correlation, however, could have arisen because the proportion of multi-brooded species was highly correlated with the incubation duty (electronic

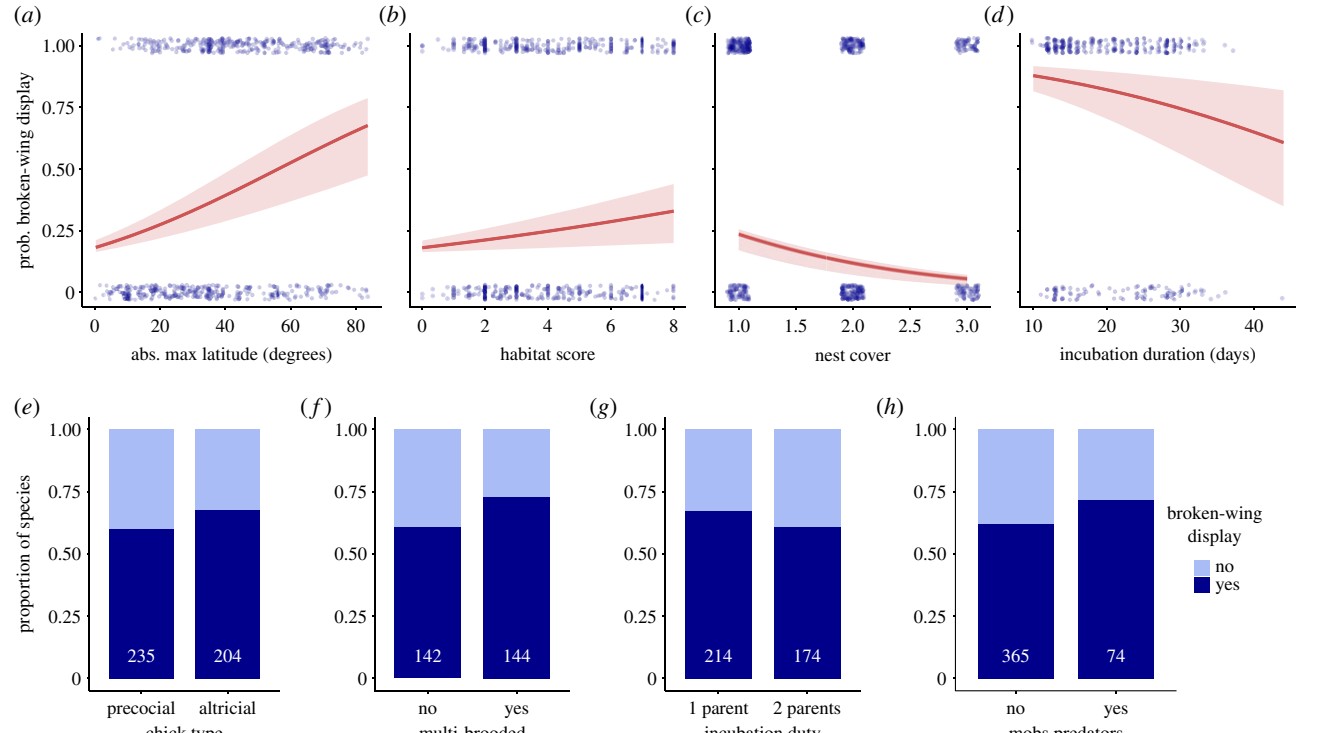

**Figure 3.** Environmental and life-history traits predict the inclination of species to perform the broken-wing display (*a–d*). Probability of species to perform the broken-wing display according to (*a*) the maximum absolute latitude of their range, (*b*) habitat score (higher values equal more forested habitat), (*c*) the amount of vegetation concealing the nest and (*d*) the incubation duration. Marginal effect predicted values (red line) are derived from a phylogenetic generalized logistic model, red shaded areas display the bootstrap confidence intervals. Raw data points are displayed in blue. (*e–h*) The proportion of species in our dataset that perform (dark blue) or do not perform (light blue) the broken-wing display according to (*e*) whether chicks are precocial or altricial, (*f*) whether the species raises several brood in one season, (*g*) whether one or both parents incubate and (*h*) whether parents mob potential predators approaching their nest. White numbers indicate the sample size for each category. (Online version in colour.)

supplementary material, figure S8), such that multi-brooded species are often species with only one parent incubating. Species maximum longevity was unrelated to the probability of performing the display (M9, $p = 0.56$, $n = 198$).

## 4. Discussion

We show that birds within 52 families and 13 orders perform the broken-wing display, indicating that it is much more widespread than previously thought [8,9]. This updated, albeit imperfect, understanding of the phylogenetic distribution of the display clearly suggests that it is not exhibited primarily by shorebirds, but widespread across Aves. Furthermore, our phylogenetically controlled models of the life-history and ecological correlates of the broken-wing display provide a needed update to our understanding of the forces that may select for this deceptive tactic and a revision of the 'type' of bird that uses this eye-catching display.

The overall picture indicates that the broken-wing display occurs in bird lineages spanning basal palaeognath species to the most derived passerines. Although our methods likely failed to identify some families with members that perform the display, we found clear clustering of the display within some clades and marked absences within others. One conspicuous absence of the trait in a large clade is in the trogons (Trogonidae) and their sister taxon (including kingfishers, rollers, woodpeckers, barbets, hornbills and hoopoes). Almost all of the birds in this taxon are cavity nesting, hinting at a possible strong association between breeding

ecology and the evolution of the broken-wing display. By and large, the frequent yet disjunct occurrence of the broken-wing display across the avian phylogenetic tree suggests that the trait has evolved several times independently. This makes the broken-wing display a paradigmatic case for the study of the ecological conditions and life-history traits associated with the evolution of a specific anti-predator behaviour.

The previous focus on shorebirds [8,9] has limited the range of proposed sources of selection that may favour the evolution of the broken-wing display (e.g. [4,7]). Our more general set of hypotheses permitted a more comprehensive investigation of the evolution of this behaviour. Of the 16 life-history and ecological traits we evaluated, eight were associated with the probability of performing the broken-wing display. We detected a strong positive correlation with the maximum absolute latitude of species' distribution and a negative correlation with nest cover, such that birds breeding at higher latitudes or with more exposed nests are more likely to perform the display than tropical birds or species with well-concealed nests. Opposite to our predictions, species from more open habitats, that have precocial young or are single-brooded are less likely to perform the display. Additionally, our analyses indicate that the broken-wing display is more common among species with shorter incubation periods. By contrast, body mass, egg mass, clutch size and mass, nest structure and location, social habits and longevity are not related to this behaviour.

To check our results for potential bias by false negatives (i.e. wrongly assigning displaying species as non-displaying because they are rarely observed and thus no report of the

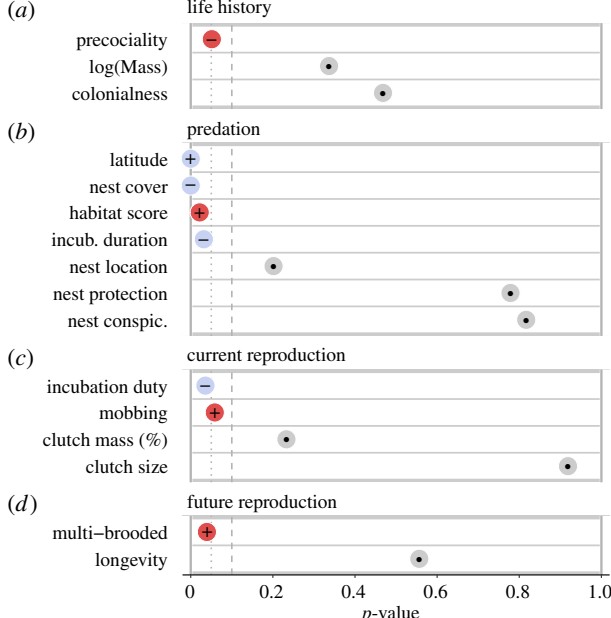

**Figure 4.** Eight of the 16 predictors tested have a significant influence on the probability of performing the broken-wing display. The *p*-value of each predictor from the simplest model with the highest sample size is displayed. The plus and minus signs show the direction of predicted effects and the colours denote whether results are consistent with the hypothesis (blue) or not (red) and grey reflects predictors that were uninformative. The vertical dotted line represents *p* = 0.05 and the dashed line represents *p* = 0.1. Predictors are organized according to their categories: traits related to life history (*a*), predation (*b*), current reproduction (*c*) and future reproduction (*d*).

display is available), we ran two additional analyses (electronic supplementary material): first, we used only temperate and polar species, which are much better studied than tropical ones. Second, we analysed only species for which the clutch size or egg measurements are known. The rationale behind this is that to be able to count the eggs, a researcher must have approached the nest, which would have elicited the broken-wing display in one or both parents, provided that the species exhibits this behaviour. Both sub-analyses largely confirmed our global analysis; except for one predictor (mobbing), they were qualitatively identical to models in the full analysis, suggesting our results are robust to any potential influence of false negatives.

Several traits related to predation risk explained variation in the occurrence of the broken-wing display, whereas only one trait related to investment in current reproduction did. Although predation risk is strongly related to reproductive strategies and other aspects of life-history variation [26,27], this contrast suggests that evolution of the broken-wing display may be more strongly shaped by differences in predation pressure or species-specific susceptibility to nest predators, rather than variation in the maximization of single reproductive attempts. Differences in predation pressure have been shown to induce short-term behavioural changes, such as activity patterns [28] or vigilance [29,30]. Our findings show how variation in predation risk can shape the evolution of behavioural anti-predator strategies. In particular, the latitudinal gradient in the probability of performing the display, originally proposed by Armstrong [7], was the strongest effect in our analysis and this effect also remained in our sensitivity analysis that excluded potentially undersampled tropical species. Latitude is related to many

ecological and environmental factors, such as day length, duration of the reproduction season and daily patterns of predation risk (e.g. night versus day). All of which could contribute to the increased probability of performing the broken-wing display with distance from the equator. Although nest predation risk declines with distance from the equator [27,31], the composition of terrestrial nest predators is thought to transition from primarily nocturnal in the tropics to diurnal in temperate areas [7] (electronic supplementary material, table S4). Because the broken-wing display is primarily targeted to visual predators, such as snakes or foxes, that would be attracted by conspicuous movements of a displaying parent, our results are consistent with the expectation that selection for the display is stronger in temperate zones.

Shorter breeding seasons could also help explain the increase in the probability of performing the broken-wing display with absolute latitude. Shorter breeding seasons are not only associated with variation in life-history traits, such as larger clutch sizes and fewer breeding attempts per season [27], but also anti-predator strategies. For example, flight initiation distance is lower at higher latitudes across bird species and correlated with lower predator abundance [32]. When breeding, cinereous tits (*Parus cinereus*) behave differently to nest predation attempts in tropical areas relative to temperate areas [33]. Tropical birds tend to abandon nests in favour of future reproductive opportunities, whereas temperate birds aggressively defend nests in favour of current brood survival. Aggressive nest defence has been studied extensively in several species [27,31,32] and increased defence intensity correlates with nest success [32]. Outwardly, aggressive nest defences, such as predator mobbing, might appear quite different from deceptive nest-defensive behaviours, such as the broken-wing display. However, our results indicate they are related, as the probability of performing the broken-wing display was significantly higher in the full dataset for mobbing species. A similar trend was observed in the sensitivity analysis considering only species with known clutch size, but the precision of the estimated effect was lower. The increase of both behaviours with absolute latitude may reflect selection for the success of the current reproductive attempt over future reproductive potential. However, we cannot rule out that some species that aggressively defend their nests were not reported, thus further investigation is necessary.

We also found the broken-wing display to be more common with decreased nest cover, which is consistent with the hypothesis that less concealed nests are more vulnerable to visual nest predators. The other measures of nest susceptibility to predation seem to be unrelated to the evolution of the broken-wing display. One potential reason for a lack of a correlation with these variables could be that the indices we used to represent nest protection and conspicuousness were based on human perceptions and not those of nest predators. Second, although Mainwaring *et al.* [34] report that a nest's crypsis predicts its probability of being preyed upon, Mouton & Martin [35] showed that the nest structure does not provide more protection against predators for altricial tropical passerines. Moreover, nest size, rather than the nest position above ground, is a more important determinant of predation [36]. In addition, nest position above ground and clumpiness within a breeding area can vary between years according to predation pressure [35,37], thus nesting habits can be quite plastic. It would be interesting to know whether the use of the broken-wing display is related to overall

behavioural plasticity, how much it varies within species and individuals, and to what degree environmental context, timing or predator type explains the variation. Some inter-specific variation in diversionary display behaviours has been documented in shorebirds: Gochfeld [8] and Armstrong [38,39] described as many as 16 categories of distraction displays, some species performing only one or two types, and other species displaying from a broader set. Future studies that characterize variation in the display within other avian families or that evaluate how the display changes within populations and individuals will reveal whether the plasticity of the display is equally widespread across Aves as is its occurrence.

Although parental investment in a brood has been proposed to explain fine-scale temporal variation in the intensity of the broken-wing display [40], we found only two of eight parental investment traits related to the display. Display intensity peaks when parental investment is the greatest and the brood is most vulnerable [40]. Such variation in the intensity of nest defence may reflect a trade-off between parental investment in the current brood versus future reproduction. Along these lines, nest defence intensity in cinereous tits [33] correlates with larger clutch size. However, our across-species analysis found no influence of parental investment on the probability to perform the broken-wing display. Of course, it is also likely that some of the predictors in our analyses coevolved with the broken-wing display as a result of increased predation pressure instead of driving the evolution of this behaviour. For example, fast nestling development is a common pattern in species that experience high nest predation [41,42]. Thus, our finding that incubation duration negatively correlates with the occurrence of the broken-wing display might reflect that both are ultimately driven by predation pressure. Similarly, some traits may belong to a suite of characters related to certain reproductive strategies instead of representing separate predictors for the evolution of the broken-wing display. This is probably the case for incubation duty and multi-brooding (electronic supplementary material, figure S9). The correlation between these predictors and the occurrence of the broken-wing display therefore might indicate an underlying role for reproductive strategies in the emergence of the broken-wing display.

As with many literature-based comparative analyses, a caveat of our study is that we could only consider species for which information has been published. We tried to mitigate this limitation by also including a survey of experts. Still, we cannot rule out that species that exhibit the broken-wing display were missed. Even though the behaviour is very obvious and can be easily spotted by human observers, there are many poorly described species throughout the world that might perform the display. Thus, our data represent a strong foundation from which to build upon observations of the broken-wing display as additional species are recorded. Future studies may further clarify the link between predation risk and the broken-wing display. For instance, as predation is thought to drive the evolution of crypsis and disruptive coloration [43], it would be interesting to investigate whether the occurrence of the broken-wing display is associated with the conspicuousness of parents or eggs.

## 5. Conclusion

We found that the broken-wing display is widespread among birds and that it has likely evolved several times independently. Latitude was the strongest predictor for the occurrence of the display, hinting at a potent effect of ecological factors and/or life-history traits that are associated with biogeography. In addition, our results suggest that the evolution of the broken-wing display is more strongly shaped by differences in predation pressure, rather than maximization of single reproductive attempts. By and large, our study not only sheds new light on a textbook example of deceptive displays, but also provides new avenues for using the broken-wing display to address fundamental questions about the evolution of anti-predator behaviour.

Data accessibility. The data supporting this study are available from the Dryad Digital Repository: https://doi.org/10.5061/dryad. fttdz08td [44].

Authors' contributions. L.F.: formal analysis, methodology, visualization, writing—original draft and writing—review and editing; W.I.T.: conceptualization, data curation, funding acquisition, investigation, methodology and writing—review and editing; S.M.D.: data curation, investigation, methodology, software and writing—review and editing; C.D.F.: conceptualization, formal analysis, funding acquisition, investigation, methodology, project administration, supervision, writing—original draft and writing—review and editing; H.B.: methodology, project administration, resources, supervision and writing—review and editing.

All authors gave final approval for publication and agreed to be held accountable for the work performed therein.

Competing interests. We declare we have no competing interests.

Funding. Open access funding provided by the Max Planck Society.

This project was supported by an Alexander von Humboldt-Stiftung Fellowship to C.D.F., a Richard A. Pimentel Fund grant to W.I.T. and the Max Planck Society to H.B.

Acknowledgements. We are indebted to survey participants for key data on the occurrence of the broken-wing display. We also thank Bart Kempenaers and three anonymous reviewers for comments on early versions of this manuscript.

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
