## [Peer Review File · Proceedings of the Royal Society B: Biological Sciences]

Review History

RSPB-2021-1850.R0 (Original submission)

Review form: Reviewer 1

Recommendation

Major revision is needed (please make suggestions in comments)

Scientific importance: Is the manuscript an original and important contribution to its field?

Acceptable

General interest: Is the paper of sufficient general interest?

Good

Quality of the paper: Is the overall quality of the paper suitable?

Acceptable

Is the length of the paper justified?

No

Should the paper be seen by a specialist statistical reviewer?

No

Do you have any concerns about statistical analyses in this paper? If so, please specify them explicitly in your report.

No

It is a condition of publication that authors make their supporting data, code and materials available - either as supplementary material or hosted in an external repository. Please rate, if applicable, the supporting data on the following criteria.

Is it accessible?

Yes

Is it clear?

Yes

Is it adequate?

Yes

Do you have any ethical concerns with this paper?

No

Comments to the Author

RSPB-2021-1850

This study looks at a variety of ecological and life history factors that correlates with the evolution of broken-wing displays in birds. This is an interesting behavior that is often featured in text-books as an iconic anti-predator strategy. I must admit that I personally quite like studies like this one – thinking about the evolution of amazing behavioral traits we have all learned about in our first animal behavior class (long ago for some of us). Thus, my own personal bias inclines me to suggest that it be published in PRSB. Regardless, I do think there are a few significant issues with the manuscript that should be addressed more thoroughly before it is published, whether that's in PRSB or elsewhere.

1. Getting to my point above, I personally like studies like this one, but I wonder about its broader significance and importance. As it's written now, it seems specifically tailored to broke-wing displays, with little justification of why we should study the evolution of this phenomenon. Don't get me wrong, I think there is a strong justification (maybe framework is a better word) for this study. For me, I think this is an incredible example of a "complex behavior" that seems to convergently evolve in a wide range of unrelated avian taxa. This has the potential to shed some interesting light on how behavioral adaptation might work. But this is just my take; there are others, and I don't want to imply that the authors must follow my point of view. Either way, I do think the authors should work to frame this study much more broadly.

2. I appreciate the long list of ecological and life history factors that the authors include in their models, and I think that their statistical approach is robust. However, I think there is confusion between factors that can sever as drives of the evolution of broken-wing displays and factors that likely do not. For example, some factors (nest cover, nest on ground) are likely signatures of selection regimes that many favor the emergence of broken-wing displays, whereas other life history variables are more likely traits that co-evolve with broke-wing displays (if at all). This should be acknowledged in the paper. In fact, I would even suggest removing the factors that likely do not drive the evolution of broken-wing displays, as I think this clouds the study and makes the results more difficult to understand. If you're aiming to test what underlies the emergence of this behavior, then do that.

3. The message of the paper is utterly lost among the laundry list of results. In my mind, the main finding is articulated on lines 307-310. That is, the results suggest that a very specific

type of predation pressure selects for broke-wing displays, as opposed to simply high or low predation pressures. Thus, there seems to be a particular behavioral solution to a specific predatory landscape that underlies the repeated emergence of broke-wing display behavior, as opposed to other anti-predator tactics. To this end, I think the authors must take a crack at articulating exactly what the predation landscape looks like that would favor the emergence of broke-wing displays. Right now, the description is hidden in the jargon of life history terminology and ecological terms.

4. Building somewhat on my point above (#3), the hypotheses about nocturnal and diurnal predation are quite weak. Shouldn't there be key citations at the end of the sentence on line 318? It would be important to run an additional analysis on a subset of the data, in which you test the idea put forth on line 320. This should be do-able and straightforward. You can probably find a handful of species in your analysis (and their close relatives) in which you can describe the main predators. If your hypothesis is true, then you should find that temperate broke-wing displaying species are more likely to experience predation from visual species than non-visual ones. One of the reasons that I think such a test is needed is that this paper becomes wishy-washy and vague about predation pressure. For example, I'm not 100% sure what you mean by visual predator (maybe a few examples would be helpful). Do you mean bird of prey vs snake? Is there data to back this up?

5. One methodological concern I have is the formation of species pairs. Is there in way to verify that the phylogenetic distance between species in a pair is the same or similar across the entire phylogeny? It doesn't seem to be a fair representation of the data if the distance between a broken-wing displaying bird and a non-displaying bird is far greater in some clades than it is on others. I think this should be addressed.

6. A final methodological concern is the survey results. I appreciate and applaud this effort, but my trust of folks' interpretations of broken-wing displays not represented in the literature is thin. I know many ornithologists who think they see birds doing things you wouldn't expect, and while I trust them, there is really no way to back this up. Is there any way to verify the robustness of the survey reports, particularly with respect to the new species that are not identified in the literature. I would, for example, feel much more comfortable if the analyses were run without these additional 61 species and the same effects were uncovered. Alternatively, could you run significant models repeatedly, but in each run you include a new subset of 20 randomly selected species from this set of 61. Again, I really like the effort made to deepen our understanding of broken-wing displays, but I'm not sure I yet believe in the veracity of these data.

Review form: Reviewer 2

Recommendation

Major revision is needed (please make suggestions in comments)

Scientific importance: Is the manuscript an original and important contribution to its field?

Acceptable

General interest: Is the paper of sufficient general interest?

Good

Quality of the paper: Is the overall quality of the paper suitable?

Acceptable

Is the length of the paper justified?

Yes

Should the paper be seen by a specialist statistical reviewer?

No

Do you have any concerns about statistical analyses in this paper? If so, please specify them explicitly in your report.

Yes

It is a condition of publication that authors make their supporting data, code and materials available - either as supplementary material or hosted in an external repository. Please rate, if applicable, the supporting data on the following criteria.

Is it accessible?

Yes

Is it clear?

Yes

Is it adequate?

Yes

Do you have any ethical concerns with this paper?

Yes

Comments to the Author

This paper seeks to provide a much more comprehensive account of the phylogenetic distribution of an avian anti-predator behavior – the avian broken wing display – than has been available to date.

It's a difficult problem because it's not easy to say definitively that a species lacks this behavior just because there is no reference to it in the literature. Furthermore, there is within species variation, as the paper points out but does not analyze. It's an interesting problem because it appears that the behavior occurs all across the avian tree of life but has blinked on and off many times during evolution. The authors here ask what aspects of life history or ecology are associated with the blinking on and off. They make a set of predictions which I found to be generally reasonable and interesting, and they test those predictions using PGLM.

I found the statistical approach to be curious because, rather than taking the entire set of bird species that are known or not known to do the display, they took each species that is known to do it, and paired it with a close relative that is not known to do it. However, they did not analyze the data as a set of pairs, as might be an effective way to ask what other changes accompany the evolutionary transition to or from using broken wing displays. Rather, they throw all 570 species (285 pairs) into one PGLM. When I started looking through the pairs, I could see perhaps why they didn't do this, as I realized that not all pairs were truly phylogenetically independent. There were very closely related pairs in some cases, such as West Peruvian Dove and White-winged Dove, and very distant ones in other cases, such as Common Nighthawk and Forbes's Plover. Looking at the species that were scored as not using the behavior, I found myself suspecting that many of those species would be discovered in the future as actually using broken wing displays (esp. oystercatcher, plover, and nightjar species); however, I did not see any cases that I could specifically refute. I believe that the literature search methods that were employed were imperfect for this purpose, but I am at least convinced that they were conducted with rigor. One general pattern that was evident was that the member of each species pair that did not use the display was frequently the one that would be expected to be more data deficient on the basis of its distribution and the amount of readily available information about its behavior and life history. The authors were aware of this shortcoming, and they at least partially addressed it by looking at the latitude effect only for high latitudes, a supplementary analysis that I found to be compelling.

The latitude result was the most convincing one of this paper in my opinion, because it implies increased nest predation pressure from mammalian predators during the elongated daylight hours of associated with high latitudes.

A major contribution of this paper was descriptive – what species use the broken wing display and what is their phylogenetic distribution? It follows that an important shortcoming of the paper, in my opinion, was not including visualization of the phylogenetic pattern of presence vs absence of the broken wing display in a way that was more detailed than the family level treatment in figure 2. The phylogenetic pattern of taxon sampling would have been very important to visualize; while I recognize that a tree with 570 tips is too large to fit a page, there are several possible ways around this, and I strongly encourage the authors to come up with one. My preferred option would be to use multiple smaller trees that occupy multiple figure panels or pages. It will be important that the readers can read the taxon names and see the transitions between display and no-display on the phylogeny (a color scheme using red and blue as the authors used in Fig. 2 would be perfect). The visual description is therefore strongly warranted even if it takes a substantial amount of space. I believe that the current figures 1 and 2 are not important to include in the main body of the paper, and I strongly encourage moving them to the supplement in order to make space for a figure that actually displays the taxon sampling scheme and phylogeny that encompass the evolutionary transitions being described and analyzed in this paper.

Specific Comments:

Phylogenetic signal is not addressed in the paper that I could find, but it's clearly relevant and can and should be reported. The models, with the pairwise taxon sampling scheme are likely to underestimate phylogenetic signal, but in the larger context of the avian tree of life, what is the degree of phylogenetic signal, and what is the estimated number of transition events between display and no-display. Figure 2 illustrates the pattern at the family level, but there is no quantification that I could find.

Line 222-223: don't assert sister taxon of turacos, as their phylogenetic placement is deeply problematic and almost certainly wrong here.

Prediction table: is the prediction and justification for double-brooding correct? Double brooders have faster life histories (more broods/year), they should also have *more*urgency to protect the present brood. (???)

Figure S3 and line 225 and possibly elsewhere: should say "significant association" not "significant influence"; generally be more careful about asserting "influence".

An additional measure to add to the ease of interpretation of this paper would be to have a single figure / table that allows comparison between the predictions (Table 1) and the results.

There was a survey of expert knowledge conducted, but none of the experts were thanked. I'm surprised at how few references there are here, as if all this information has been effectively purged of its original sourcing. Who else should be acknowledged and cited here? I encourage the authors to consider this deeply before the next stage of publication.

Review form: Reviewer 3

Recommendation

Major revision is needed (please make suggestions in comments)

Scientific importance: Is the manuscript an original and important contribution to its field?

Good

General interest: Is the paper of sufficient general interest?

Good

Quality of the paper: Is the overall quality of the paper suitable?

Acceptable

Is the length of the paper justified?

Yes

Should the paper be seen by a specialist statistical reviewer?

Yes

Do you have any concerns about statistical analyses in this paper? If so, please specify them explicitly in your report.

Yes

It is a condition of publication that authors make their supporting data, code and materials available - either as supplementary material or hosted in an external repository. Please rate, if applicable, the supporting data on the following criteria.

Is it accessible?

Yes

Is it clear?

Yes

Is it adequate?

Yes

Do you have any ethical concerns with this paper?

No

Comments to the Author

This manuscript explores the evolution of the broken-wing display, a fascinating distraction and anti-predation behavior that is unique in its complexity and ability to captivate human observers. The authors first identify species that display the trait by using literature searches and surveys, and show that it has likely evolved several times outside the shorebird clades, which are classically associated with the behavior. The manuscript then reviews 16 proposed hypotheses for the evolution of the display, and test predictions using phylogenetically-informed models. The considerable effort to analyze the extent of its evolution is impressive. The manuscript shows that there is still a great deal to be learned about the repeated evolution of this complex trait.

As someone who studies bird behavior myself, I found the study of this trait to be interesting. However, I think the authors could make a much better attempt to persuade a wider audience as to why this trait and its evolution should be of interest. Currently the main justification seems to be that the display is well-known and "textbook", but this strikes me as a somewhat avian-centric claim. It would help to better use the introduction to place the broken-wing display among other types of deceptive or distractive behaviors. Do non-avian animals use any similar distraction displays during parental care? Why should the audience care about this behavior when it is one of many types of anti-predator strategies used by birds?

Another general concern I have is regarding the high likelihood of false negatives in this study and how that might affect the results. As the authors note in the discussion, many species that exhibit broken-wing displays have likely not been reported. What I believe is still missing, however, is a consideration for how species conspicuousness and rarity will affect the results.

This is especially the case if a common or conspicuous species that exhibits the display is “paired” (line 102) with a “no-display” species that is rare, inconspicuous, or has a small range that would limit observation and increase the probability of a false negative. At the very least, it would be nice to have more discussion on how these issues might bias the results and in which directions. However, one potential way to make these results more convincing would be to include a parameter(s) that is directly related to species visibility to humans. For example, the annual sighting rate of each species on ebird would give a good approximation for how often visible species are (and therefore able to be accurately reported). Including this parameter in the models would allow for the model to account for the issue of conspicuousness and reporting problems.

I have also listed line-by-line comments below, most of which are in regards to much-needed clarifications and justifications for various methods.

Lines 90-97: Since the survey added a significant number of species to the dataset, I find myself wanting to know a lot more about the methodology here. Did the authors simply ask participants to specify species that they had seen do a broken-wing display? Were participants told to exclude species that had already been found in the literature review? What is meant by “environmental context of the observation” (line 91) and how was this information used? How was an observation deemed trustworthy? Were any observations rejected and if so, why? The survey wording itself would be valuable information to include in the supplementary materials. Which internationally-known ornithological organizations were contacted? An analysis of the international distribution of responses would also be informative since biased reporting could play a major role in the results.

Line 142: I’m not sure I understand how scrape nests are considered both inconspicuous and also non-protected at the same time.

Line 169-170: Assuming no report of aggressive nest defense reflects an absence of the trait requires us to assume equal observation rate. Is that a safe assumption?

Lines 186-208: This is an intriguing model selection technique that I have not personally seen before. I do not have the expertise to comprehensively judge its competence. Regardless, I believe a great deal of clarification is needed. I do not see any citation to statistical papers or other phylogenetic papers that use a similar technique, so I am unsure how to consider its utility and appropriateness. I understand that the authors were dealing with a variably complete dataset, but this does not clearly justify the method. Other questions I am left with:

- Why did the authors start with only 4 variables? Is there a reason to choose 4?
- Why was only one variable added to make model M1, while it seems 4 variables were added to make model M2? It seems that in most cases variables were added back until 4 variables were reached again, so what’s going on with M2?
- It might help to bold the variables in the “full” models that were $p < 0.1$.
- Line 197-199: Removing an important predictor variable is almost certain to inflate the importance of variables with smaller effect sizes.
- Line 199: It seems to me that collinearity should be checked with the full model before simplification because non-collinearity can lead to incorrect model conclusions.
- On the same topic, how can the authors be sure collinearity among variables is not an issue? The technique used here never considers a truly full model with all 16 explanatory variables. The results will almost certainly inflate the number of significantly predictive variables because two correlated variables that are never considered in the same model will both appear to have explanatory power when considered separately.

Line 243: Given the very high likelihood that this variable is skewed by biased reporting in these regions, I personally am not very convinced by this sensitivity analysis. Why was 30 chosen?

Could the result be spurious? What if a limit of 20, 40, 50, etc. were chosen?

Supplement:

Line 24: Why use a summarized description rather than the full HBW habitat description? How exactly was a summary made? How is a one-word summary possible?

Decision letter (RSPB-2021-1850.R0)

12-Oct-2021

Dear Dr Francis:

I am writing to inform you that your manuscript RSPB-2021-1850 entitled "The broken-wing display across birds and the conditions for its evolution" has, in its current form, been rejected for publication in Proceedings B.

This action has been taken on the advice of referees, who have recommended that substantial revisions are necessary. With this in mind we would be happy to consider a resubmission, provided the comments of the referees are fully addressed. However please note that this is not a provisional acceptance.

Sincerely,

Dr Maurine Neiman

Associate Editor

Board Member: 1

Comments to Author:

This study investigates the evolution of the broken-wing display which is an interesting and puzzling and eye-catching behavior. This work has thus the potential to be of interest to a broad audience. Unfortunately, at this point, the manuscript falls short and does not provide a strong case for this work that would make it attractive to a general audience. The study should establish the broader significance of this work beyond how attractive the behavior is on the surface.

In addition, several important methodological concerns were raised by the reviewers. Of those, two deserve critical attention. On one hand, there is the issue of false negatives due to the challenge of knowing whether a species truly does not perform the broken-wing display. While the authors recognize this shortcoming, there is no critical discussion about potential implications and how this issue could affect the findings presented here. An additional, and related, concern revolves around the statistical approach chosen. The data is analyzed by pairing species with and without the behavior which unnecessarily reduces the amount of information to the chosen species pairs. In addition, this approach weights more heavily a few species selected by the authors for which the behavior has not been reported potentially inflating the effect of false negatives in those species.

The reviewers provide insightful comments and detailed suggestions that will help improve this work. This study addresses a neat question and has the potential to be an impactful study in behavioral ecology, I hope the authors will appreciate the feedback and use it to further improve it.

Reviewer(s)' Comments to Author:

Referee: 1

Comments to the Author(s)

RSPB-2021-1850

This study looks at a variety of ecological and life history factors that correlates with the evolution of broken-wing displays in birds. This is an interesting behavior that is often featured in text-books as an iconic anti-predator strategy. I must admit that I personally quite like studies like this one – thinking about the evolution of amazing behavioral traits we have all learned about in our first animal behavior class (long ago for some of us). Thus, my own personal bias inclines me to suggest that it be published in PRSB. Regardless, I do think there are a few significant issues with the manuscript that should be addressed more thoroughly before it is published, whether that's in PRSB or elsewhere.

1. Getting to my point above, I personally like studies like this one, but I wonder about its broader significance and importance. As it's written now, it seems specifically tailored to broken-wing displays, with little justification of why we should study the evolution of this phenomenon. Don't get me wrong, I think there is a strong justification (maybe framework is a better word) for this study. For me, I think this is an incredible example of a "complex behavior" that seems to convergently evolve in a wide range of unrelated avian taxa. This has the potential to shed some interesting light on how behavioral adaptation might work. But this is just my take; there are others, and I don't want to imply that the authors must follow my point of view. Either way, I do think the authors should work to frame this study much more broadly.

2. I appreciate the long list of ecological and life history factors that the authors include in their models, and I think that their statistical approach is robust. However, I think there is confusion between factors that can serve as drivers of the evolution of broken-wing displays and factors that likely do not. For example, some factors (nest cover, nest on ground) are likely signatures of selection regimes that many favor the emergence of broken-wing displays, whereas other life history variables are more likely traits that co-evolve with broken-wing displays (if at all). This should be acknowledged in the paper. In fact, I would even suggest removing the factors that

likely do not drive the evolution of broken-wing displays, as I think this clouds the study and makes the results more difficult to understand. If you're aiming to test what underlies the emergence of this behavior, then do that.

3. The message of the paper is utterly lost among the laundry list of results. In my mind, the main finding is articulated on lines 307-310. That is, the results suggest that a very specific type of predation pressure selects for broke-wing displays, as opposed to simply high or low predation pressures. Thus, there seems to be a particular behavioral solution to a specific predatory landscape that underlies the repeated emergence of broke-wing display behavior, as opposed to other anti-predator tactics. To this end, I think the authors must take a crack at articulating exactly what the predation landscape looks like that would favor the emergence of broke-wing displays. Right now, the description is hidden in the jargon of life history terminology and ecological terms.

4. Building somewhat on my point above (#3), the hypotheses about nocturnal and diurnal predation are quite weak. Shouldn't there be key citations at the end of the sentence on line 318? It would be important to run an additional analysis on a subset of the data, in which you test the idea put forth on line 320. This should be do-able and straightforward. You can probably find a handful of species in your analysis (and their close relatives) in which you can describe the main predators. If your hypothesis is true, then you should find that temperate broke-wing displaying species are more likely to experience predation from visual species than non-visual ones. One of the reasons that I think such a test is needed is that this paper becomes wishy-washy and vague about predation pressure. For example, I'm not 100% sure what you mean by visual predator (maybe a few examples would be helpful). Do you mean bird of prey vs snake? Is there data to back this up?

5. One methodological concern I have is the formation of species pairs. Is there in way to verify that the phylogenetic distance between species in a pair is the same or similar across the entire phylogeny? It doesn't seem to be a fair representation of the data if the distance between a broken-wing displaying bird and a non-displaying bird is far greater in some clades than it is on others. I think this should be addressed.

6. A final methodological concern is the survey results. I appreciate and applaud this effort, but my trust of folks' interpretations of broken-wing displays not represented in the literature is thin. I know many ornithologists who think they see birds doing things you wouldn't expect, and while I trust them, there is really no way to back this up. Is there any way to verify the robustness of the survey reports, particularly with respect to the new species that are not identified in the literature. I would, for example, feel much more comfortable if the analyses were run without these additional 61 species and the same effects were uncovered. Alternatively, could you run significant models repeatedly, but in each run you include a new subset of 20 randomly selected species from this set of 61. Again, I really like the effort made to deepen our understanding of broken-wing displays, but I'm not sure I yet believe in the veracity of these data.

Referee: 2

Comments to the Author(s)

This paper seeks to provide a much more comprehensive account of the phylogenetic distribution of an avian anti-predator behavior – the avian broken wing display – than has been available to date.

It's a difficult problem because it's not easy to say definitively that a species lacks this behavior just because there is no reference to it in the literature. Furthermore, there is within species variation, as the paper points out but does not analyze. It's an interesting problem because it appears that the behavior occurs all across the avian tree of life but has blinked on and off many times during evolution. The authors here ask what aspects of life history or ecology are associated with the blinking on and off. They make a set of predictions which I found to be generally reasonable and interesting, and they test those predictions using PGLM.

I found the statistical approach to be curious because, rather than taking the entire set of bird species that are known or not known to do the display, they took each species that is known to do it, and paired it with a close relative that is not known to do it. However, they did not analyze the data as a set of pairs, as might be an effective way to ask what other changes accompany the evolutionary transition to or from using broken wing displays. Rather, they throw all 570 species (285 pairs) into one PGLM. When I started looking through the pairs, I could see perhaps why they didn't do this, as I realized that not all pairs were truly phylogenetically independent. There were very closely related pairs in some cases, such as West Peruvian Dove and White-winged Dove, and very distant ones in other cases, such as Common Nighthawk and Forbes's Plover. Looking at the species that were scored as not using the behavior, I found myself suspecting that many of those species would be discovered in the future as actually using broken wing displays (esp. oystercatcher, plover, and nightjar species); however, I did not see any cases that I could specifically refute. I believe that the literature search methods that were employed were imperfect for this purpose, but I am at least convinced that they were conducted with rigor. One general pattern that was evident was that the member of each species pair that did not use the display was frequently the one that would be expected to be more data deficient on the basis of its distribution and the amount of readily available information about its behavior and life history. The authors were aware of this shortcoming, and they at least partially addressed it by looking at the latitude effect only for high latitudes, a supplementary analysis that I found to be compelling. The latitude result was the most convincing one of this paper in my opinion, because it implies increased nest predation pressure from mammalian predators during the elongated daylight hours of associated with high latitudes.

A major contribution of this paper was descriptive - what species use the broken wing display and what is their phylogenetic distribution? It follows that an important shortcoming of the paper, in my opinion, was not including visualization of the phylogenetic pattern of presence vs absence of the broken wing display in a way that was more detailed than the family level treatment in figure 2. The phylogenetic pattern of taxon sampling would have been very important to visualize; while I recognize that a tree with 570 tips is too large to fit a page, there are several possible ways around this, and I strongly encourage the authors to come up with one. My preferred option would be to use multiple smaller trees that occupy multiple figure panels or pages. It will be important that the readers can read the taxon names and see the transitions between display and no-display on the phylogeny (a color scheme using red and blue as the authors used in Fig. 2 would be perfect). The visual description is therefore strongly warranted even if it takes a substantial amount of space. I believe that the current figures 1 and 2 are not important to include in the main body of the paper, and I strongly encourage moving them to the supplement in order to make space for a figure that actually displays the taxon sampling scheme and phylogeny that encompass the evolutionary transitions being described and analyzed in this paper.

Specific Comments:

Phylogenetic signal is not addressed in the paper that I could find, but it's clearly relevant and can and should be reported. The models, with the pairwise taxon sampling scheme are likely to underestimate phylogenetic signal, but in the larger context of the avian tree of life, what is the degree of phylogenetic signal, and what is the estimated number of transition events between display and no-display. Figure 2 illustrates the pattern at the family level, but there is no quantification that I could find.

Line 222-223: don't assert sister taxon of turacos, as their phylogenetic placement is deeply problematic and almost certainly wrong here.

Prediction table: is the prediction and justification for double-brooding correct? Double brooders have faster life histories (more broods/year), they should also have *more*urgency to protect the present brood. (???)

Figure S3 and line 225 and possibly elsewhere: should say “significant association” not “significant influence”; generally be more careful about asserting “influence”.

An additional measure to add to the ease of interpretation of this paper would be to have a single figure / table that allows comparison between the predictions (Table 1) and the results.

There was a survey of expert knowledge conducted, but none of the experts were thanked. I'm surprised at how few references there are here, as if all this information has been effectively purged of its original sourcing. Who else should be acknowledged and cited here? I encourage the authors to consider this deeply before the next stage of publication.

Referee: 3

Comments to the Author(s)

This manuscript explores the evolution of the broken-wing display, a fascinating distraction and anti-predation behavior that is unique in its complexity and ability to captivate human observers. The authors first identify species that display the trait by using literature searches and surveys, and show that it has likely evolved several times outside the shorebird clades, which are classically associated with the behavior. The manuscript then reviews 16 proposed hypotheses for the evolution of the display, and test predictions using phylogenetically-informed models. The considerable effort to analyze the extent of its evolution is impressive. The manuscript shows that there is still a great deal to be learned about the repeated evolution of this complex trait.

As someone who studies bird behavior myself, I found the study of this trait to be interesting. However, I think the authors could make a much better attempt to persuade a wider audience as to why this trait and its evolution should be of interest. Currently the main justification seems to be that the display is well-known and “textbook”, but this strikes me as a somewhat avian-centric claim. It would help to better use the introduction to place the broken-wing display among other types of deceptive or distractive behaviors. Do non-avian animals use any similar distraction displays during parental care? Why should the audience care about this behavior when it is one of many types of anti-predator strategies used by birds?

Another general concern I have is regarding the high likelihood of false negatives in this study and how that might affect the results. As the authors note in the discussion, many species that exhibit broken-wing displays have likely not been reported. What I believe is still missing, however, is a consideration for how species conspicuousness and rarity will affect the results. This is especially the case if a common or conspicuous species that exhibits the display is “paired” (line 102) with a “no-display” species that is rare, inconspicuous, or has a small range that would limit observation and increase the probability of a false negative. At the very least, it would be nice to have more discussion on how these issues might bias the results and in which directions. However, one potential way to make these results more convincing would be to include a parameter(s) that is directly related to species visibility to humans. For example, the annual sighting rate of each species on ebird would give a good approximation for how often visible species are (and therefore able to be accurately reported). Including this parameter in the models would allow for the model to account for the issue of conspicuousness and reporting problems.

I have also listed line-by-line comments below, most of which are in regards to much-needed clarifications and justifications for various methods.

Lines 90-97: Since the survey added a significant number of species to the dataset, I find myself wanting to know a lot more about the methodology here. Did the authors simply ask participants to specify species that they had seen do a broken-wing display? Were participants told to exclude species that had already been found in the literature review? What is meant by “environmental context of the observation” (line 91) and how was this information used? How was an

observation deemed trustworthy? Were any observations rejected and if so, why? The survey wording itself would be valuable information to include in the supplementary materials. Which internationally-known ornithological organizations were contacted? An analysis of the international distribution of responses would also be informative since biased reporting could play a major role in the results.

Line 142: I'm not sure I understand how scrape nests are considered both inconspicuous and also non-protected at the same time.

Line 169-170: Assuming no report of aggressive nest defense reflects an absence of the trait requires us to assume equal observation rate. Is that a safe assumption?

Lines 186-208: This is an intriguing model selection technique that I have not personally seen before. I do not have the expertise to comprehensively judge its competence. Regardless, I believe a great deal of clarification is needed. I do not see any citation to statistical papers or other phylogenetic papers that use a similar technique, so I am unsure how to consider its utility and appropriateness. I understand that the authors were dealing with a variably complete dataset, but this does not clearly justify the method. Other questions I am left with:

-- Why did the authors start with only 4 variables? Is there a reason to choose 4?

-- Why was only one variable added to make model M1, while it seems 4 variables were added to make model M2? It seems that in most cases variables were added back until 4 variables were reached again, so what's going on with M2?

-- It might help to bold the variables in the "full" models that were $p < 0.1$.

-- Line 197-199: Removing an important predictor variable is almost certain to inflate the importance of variables with smaller effect sizes.

-- Line 199: It seems to me that collinearity should be checked with the full model before simplification because non-collinearity can lead to incorrect model conclusions.

-- On the same topic, how can the authors be sure collinearity among variables is not an issue? The technique used here never considers a truly full model with all 16 explanatory variables. The results will almost certainly inflate the number of significantly predictive variables because two correlated variables that are never considered in the same model will both appear to have explanatory power when considered separately.

Line 243: Given the very high likelihood that this variable is skewed by biased reporting in these regions, I personally am not very convinced by this sensitivity analysis. Why was 30 chosen? Could the result be spurious? What if a limit of 20, 40, 50, etc. were chosen?

Supplement:

Line 24: Why use a summarized description rather than the full HBW habitat description? How exactly was a summary made? How is a one-word summary possible?

Author's Response to Decision Letter for (RSPB-2021-1850.R0)

See Appendix A.

RSPB-2022-0058.R0

Review form: Reviewer 1

Recommendation

Accept as is

Scientific importance: Is the manuscript an original and important contribution to its field?

Good

General interest: Is the paper of sufficient general interest?

Good

Quality of the paper: Is the overall quality of the paper suitable?

Good

Is the length of the paper justified?

Yes

Should the paper be seen by a specialist statistical reviewer?

No

Do you have any concerns about statistical analyses in this paper? If so, please specify them explicitly in your report.

No

It is a condition of publication that authors make their supporting data, code and materials available - either as supplementary material or hosted in an external repository. Please rate, if applicable, the supporting data on the following criteria.

Is it accessible?

Yes

Is it clear?

Yes

Is it adequate?

Yes

Do you have any ethical concerns with this paper?

No

Comments to the Author

The authors make a strong effort to improve the manuscript, and they are successful in many ways. While the same issues still weaken the paper, they are more thoroughly addressed and acknowledged in this version of the manuscript. Hope to see it in print soon!

Review form: Reviewer 2

Recommendation

Accept with minor revision (please list in comments)

Scientific importance: Is the manuscript an original and important contribution to its field?

Good

General interest: Is the paper of sufficient general interest?

Acceptable

Quality of the paper: Is the overall quality of the paper suitable?

Good

Is the length of the paper justified?

Yes

Should the paper be seen by a specialist statistical reviewer?

No

Do you have any concerns about statistical analyses in this paper? If so, please specify them explicitly in your report.

No

It is a condition of publication that authors make their supporting data, code and materials available - either as supplementary material or hosted in an external repository. Please rate, if applicable, the supporting data on the following criteria.

Is it accessible?

Yes

Is it clear?

Yes

Is it adequate?

Yes

Do you have any ethical concerns with this paper?

No

Comments to the Author

I was pleased to see the improvements and clarifications to this. I think the phylogeny supplemental figures are helpful, even if not as informative as I had hoped (I'll just note that they look like time-trees but there are no time-scales provided). I would suggest that Fig. S3 (a nice addition!) become Fig. 4 of the main paper. However, there was a weird design decision with Fig. S3 that authors should consider. It took me a few minutes to realize how to interpret it because I had assumed that the +/- symbols would correspond to the direction of the *effect* (which I think is more logical way to read this) when they actually corresponded to the direction of the *hypothesis*. To figure out the direction of the effect, the reader needs to consider both the symbol and the color (a bit convoluted). Anyway, that's a subjective thing, but I'll leave it to the authors to consider changing it. Regarding the issue about our disagreement over a priori expectations around double-brooding and fast life history: my logic related to the tradeoff between parental investment in the current brood versus future ones: double brooding species should tend to have faster life histories in general (lower survival, faster reproduction), and therefore their individual fitness could be more affected by loss of a single brood (whereas a species with a slower life history would survive to breed again the following year); it follows that, for double-brooders, expenditure of energy and risk to avoid loss of a clutch would be more likely to benefit fitness, therefore broken-wing display would be more likely to evolve.

The Dryad data and metadata look excellent and complete.

Decision letter (RSPB-2022-0058.R0)

14-Feb-2022

Dear Dr Francis

I am pleased to inform you that your manuscript RSPB-2022-0058 entitled "The broken-wing display across birds and the conditions for its evolution" has been accepted for publication in Proceedings B.

The referee(s) have recommended publication, but also suggest some minor revisions to your manuscript. Therefore, I invite you to respond to the referee(s)' comments and revise your manuscript. Because the schedule for publication is very tight, it is a condition of publication that you submit the revised version of your manuscript within 7 days. If you do not think you will be able to meet this date please let us know.

Sincerely,

Dr Maurine Neiman

Associate Editor

Comments to Author:

I commend the authors on successfully addressing the concerns brought up in the review process. As both reviewers recognized, the manuscript has improved tremendously. The addition of the sensitivity analysis, in particular, has helped make a strong case. Overall, the study reads well and provides a more robust exploration of the potential factors associated with the evolution of this interesting behavior.

I agree with reviewer 2 that Fig. S3 is a great addition and would be a nice complement to the figures already in the main text. However, in its current form it is really difficult to unpack the information it presents. My concern is that in this figure the reader can easily get a misleading impression of the data. The reviewer provides an excellent description of the main issue of this figure and I strongly recommend re-structuring it to better display the data and support (or lack of support) for the hypotheses. I look forward to seeing the revised version of this figure.

Reviewer(s)' Comments to Author:

Referee: 1

Comments to the Author(s).

The authors make a strong effort to improve the manuscript, and they are successful in many ways. While the same issues still weaken the paper, they are more thoroughly addressed and acknowledged in this version of the manuscript. Hope to see it in print soon!

Referee: 2

Comments to the Author(s).

I was pleased to see the improvements and clarifications to this. I think the phylogeny supplemental figures are helpful, even if not as informative as I had hoped (I'll just note that they look like time-trees but there are no time-scales provided). I would suggest that Fig. S3 (a nice addition!) become Fig. 4 of the main paper. However, there was a weird design decision with Fig. S3 that authors should consider. It took me a few minutes to realize how to interpret it because I had assumed that the +/- symbols would correspond to the direction of the *effect* (which I think is more logical way to read this) when they actually corresponded to the direction of the *hypothesis*. To figure out the direction of the effect, the reader needs to consider both the symbol and the color (a bit convoluted). Anyway, that's a subjective thing, but I'll leave it to the authors to consider changing it. Regarding the issue about our disagreement over a priori expectations around double-brooding and fast life history: my logic related to the tradeoff between parental investment in the current brood versus future ones: double brooding species should tend to have faster life histories in general (lower survival, faster reproduction), and therefore their individual fitness could be more affected by loss of a single brood (whereas a species with a slower life history would survive to breed again the following year); it follows that, for double-brooders, expenditure of energy and risk to avoid loss of a clutch would be more likely to benefit fitness, therefore broken-wing display would be more likely to evolve.

The Dryad data and metadata look excellent and complete.

Author's Response to Decision Letter for (RSPB-2022-0058.R0)

See Appendix B.

Decision letter (RSPB-2022-0058.R1)

28-Feb-2022

Dear Dr Francis

I am pleased to inform you that your manuscript entitled "The broken-wing display across birds and the conditions for its evolution" has been accepted for publication in Proceedings B.

Data Accessibility section

Open Access

Paper charges

Sincerely,

Proceedings B

Appendix A

RSPB-2021-1850 RESPONSE TO EDITOR AND REVIEWER COMMENTS

Associate Editor

This study investigates the evolution of the broken-wing display which is an interesting and puzzling and eye-catching behavior. This work has thus the potential to be of interest to a broad audience. Unfortunately, at this point, the manuscript falls short and does not provide a strong case for this work that would make it attractive to a general audience. The study should establish the broader significance of this work beyond how attractive the behavior is on the surface.

In addition, several important methodological concerns were raised by the reviewers. Of those, two deserve critical attention. On one hand, there is the issue of false negatives due to the challenge of knowing whether a species truly does not perform the broken-wing display. While the authors recognize this shortcoming, there is no critical discussion about potential implications and how this issue could affect the findings presented here. An additional, and related, concern revolves around the statistical approach chosen. The data is analyzed by pairing species with and without the behavior which unnecessarily reduces the amount of information to the chosen species pairs. In addition, this approach weights more heavily a few species selected by the authors for which the behavior has not been reported potentially inflating the effect of false negatives in those species.

The reviewers provide insightful comments and detailed suggestions that will help improve this work. This study addresses a neat question and has the potential to be an impactful study in behavioral ecology, I hope the authors will appreciate the feedback and use it to further improve it.

Response: We were happy to learn that the associate editor found our research to be potentially impactful to behavioral ecology and also attractive to a broader audience. We have revised the introduction to better place our study in a broader context. Additionally, we have clarified our approach and completed a series of sensitivity analyses to address the concerns raised by reviewers and highlighted by the associate editor. We are happy to report that the additional analyses, which we now report in the supplement, match those reported in the main manuscript in terms of the directionality of the effects of continuous variables and the differences between categorical variables. We also have expanded the discussion to more thoroughly address the issue of false negatives, although we believe our sub-analyses restricted to species with temperate ranges and to those where clutch size is known provides additional confidence in our results. Additionally, we have clarified how we chose species with no information about performing the display. Our prior description of “pairing” species was misleading and does not adequately reflect the approach we used to balance observations as much as possible across the tree. We hope the revised text makes this clear. Finally, it is worth considering the potential tension between the critique of false negatives and the critique of choosing too few species that appear not to exhibit the display. Specifically, the associate editor suggests our approach “unnecessarily reduces the amount of information to the chosen species pairs.” This seems to imply that we

might consider increasing the number of species that appear not to exhibit the display in our dataset, but this approach would likely further increase the potential influence of false negatives. Thus, we fully agree that the two main critiques are potential drawbacks of the study and we have spent considerable time exploring alternative approaches. Because the three reviewers and associate editor did not suggest any alternative approaches, we feel our approach, even with the explicitly acknowledged limitations, is among the best ways to balance data availability, representation across the phylogeny and imperfect information on the occurrence of the display.

Reviewer(s)' Comments to Author:

Referee: 1

Comments to the Author(s)

This study looks at a variety of ecological and life history factors that correlates with the evolution of broken-wing displays in birds. This is an interesting behavior that is often featured in text-books as an iconic anti-predator strategy. I must admit that I personally quite like studies like this one—thinking about the evolution of amazing behavioral traits we have all learned about in our first animal behavior class (long ago for some of us). Thus, my own personal bias inclines me to suggest that it be published in PRSB. Regardless, I do think there are a few significant issues with the manuscript that should be addressed more thoroughly before it is published, whether that's in PRSB or elsewhere.

1. Getting to my point above, I personally like studies like this one, but I wonder about its broader significance and importance. As it's written now, it seems specifically tailored to broke-wing displays, with little justification of why we should study the evolution of this phenomenon. Don't get me wrong, I think there is a strong justification (maybe framework is a better word) for this study. For me, I think this is an incredible example of a “complex behavior” that seems to convergently evolve in a wide range of unrelated avian taxa. This has the potential to shed some interesting light on how behavioral adaptation might work. But this is just my take; there are others, and I don't want to imply that the authors must follow my point of view. Either way, I do think the authors should work to frame this study much more broadly.

Response: This comment made us realize that we focused too much on the display itself when framing our study – it seems we got carried away by the peculiar behaviour and neglected to highlight the broader implications of our work. In the light of this comment (and the similar comments by Reviewer 3 and the Associate Editor), we have revised the abstract and the introduction and now we emphasize more clearly the significance of our study regarding behavioural adaptation in general and the evolution of anti-predator behaviour in particular (lines 25-27, 34-35, 69-74).

2. I appreciate the long list of ecological and life history factors that the authors include in their models, and I think that their statistical approach is robust. However, I think there is confusion

between factors that can serve as drivers of the evolution of broken-wing displays and factors that likely do not. For example, some factors (nest cover, nest on ground) are likely signatures of selection regimes that many favor the emergence of broken-wing displays, whereas other life history variables are more likely traits that co-evolve with broken-wing displays (if at all). This should be acknowledged in the paper. In fact, I would even suggest removing the factors that likely do not drive the evolution of broken-wing displays, as I think this clouds the study and makes the results more difficult to understand. If you're aiming to test what underlies the emergence of this behavior, then do that.

Response: We completely agree with the reviewer that some of the identified factors may have co-evolved with the display and thus do not drive the evolution of the display itself. However, it seems to us that a clear distinction between the categories isn't always possible. To avoid misunderstandings, we are now using a more cautious wording, saying that a factor is "associated with the evolution of the display" instead of "driving the evolution of the display" because the reviewer is quite correct that many of these variables may co-evolve with the display and not necessarily cause evolution of the display.

We also agree that omitting some predictor variables would make the results more straightforward. On the other hand, we feel uncomfortable doing this because we know the effects and p-values for each predictor, and we are worried that a reduction of variables in hindsight may bias our analyses.

3. The message of the paper is utterly lost among the laundry list of results. In my mind, the main finding is articulated on lines 307-310. That is, the results suggest that a very specific type of predation pressure selects for broken-wing displays, as opposed to simply high or low predation pressures. Thus, there seems to be a particular behavioral solution to a specific predatory landscape that underlies the repeated emergence of broken-wing display behavior, as opposed to other anti-predator tactics. To this end, I think the authors must take a crack at articulating exactly what the predation landscape looks like that would favor the emergence of broken-wing displays. Right now, the description is hidden in the jargon of life history terminology and ecological terms.

Response: We appreciate the comment; indeed we have included many variables in our analysis. As one can see from table 1, many of these variables are associated with hypotheses in the literature about the forces that promote the occurrence of the broken-wing display, and therefore justify quantitative testing here. We tried to organize the results section in themes, thinking that this would make the findings clear to the reader.

Upon reading this comment, we tried several reorganizations of the discussion to more clearly, and earlier in the discussion, highlight the importance of predation pressure, but in all attempts we felt that the discussion did not flow well, thus we have opted to keep the more specific discussion of predation approximately where it was in the discussion previously. Still, we feel the other changes that we have made to the discussion should help clarify the main findings.

Finally, in terms of the predatory landscape, we fully acknowledge that this is a very interesting component that could influence anti-predator behavior, but that the scale of the predatory landscape is at the population or sub-population level rather than the species level; as the suite

of nest predators will invariably vary in relative abundance across any species' breeding range. Nevertheless, we have gone to great efforts to better define how predation by diurnal and nocturnal terrestrial predators varies with latitude and hence can represent a coarse picture of how the predatory landscape varies temporally. This supplementary analysis provides empirical support for Armstrong's (1954) assumption that predation by diurnal predators increases with distance from the equator and support for the hypothesis that this pattern may help explain the positive relationship between absolute latitude and probability of the occurrence of the display.

4. Building somewhat on my point above (#3), the hypotheses about nocturnal and diurnal predation are quite weak. Shouldn't there be key citations at the end of the sentence on line 318? It would be important to run an additional analysis on a subset of the data, in which you test the idea put forth on line 320. This should be do-able and straightforward. You can probably find a handful of species in your analysis (and their close relatives) in which you can describe the main predators. If your hypothesis is true, then you should find that temperate broken-wing displaying species are more likely to experience predation from visual species than non-visual ones. One of the reasons that I think such a test is needed is that this paper becomes wishy-washy and vague about predation pressure. For example, I'm not 100% sure what you mean by visual predator (maybe a few examples would be helpful). Do you mean bird of prey vs snake? Is there data to back this up?

Response: We appreciate the suggestion to pinpoint predation more specifically and have adopted the reviewer's suggestion to run some additional analyses. As described in our response to the last comment, we now provide an analysis of how nest predation by diurnal and nocturnal predators varies with latitude. We provide the full details in the supplement. Briefly, we analyzed predation events of 2863 nests by terrestrial predators reported in 110 studies that span the tropics to polar regions. We hope the reviewer will now find the link between occurrence of the broken-wing display to predation by diurnal predators more convincing.

5. One methodological concern I have is the formation of species pairs. Is there in way to verify that the phylogenetic distance between species in a pair is the same or similar across the entire phylogeny? It doesn't seem to be a fair representation of the data if the distance between a broken-wing displaying bird and a non-displaying bird is far greater in some clades than it is on others. I think this should be addressed.

Response: First, as we mention in the response to the associate editor's comments, we recognize that species pairs was poor wording. We did not use a paired design, rather, we selected related non-displaying species in order to balance the number of displaying and non-displaying species in the analysis (also see below).

Even with the effort to balance species across represented clades, we are aware that our sampling method is imperfect, but varying phylogenetic distances between displaying and non-

displaying species in some clades is what evolution provides us to work with. However, we believe this is no cause for concern because including the phylogenetic tree in the model explicitly accounts for the varying degrees of relatedness among all species in the analyses. We additionally allowed the model to estimate the phylogenetic signal in the relationship between the response variable and predictors (this is alpha for the phyloglm function).

6. A final methodological concern is the survey results. I appreciate and applaud this effort, but my trust of folks' interpretations of broken-wing displays not represented in the literature is thin. I know many ornithologists who think they see birds doing things you wouldn't expect, and while I trust them, there is really no way to back this up. Is there any way to verify the robustness of the survey reports, particularly with respect to the new species that are not identified in the literature. I would, for example, feel much more comfortable if the analyses were run without these additional 61 species and the same effects were uncovered. Alternatively, could you run significant models repeatedly, but in each run you include a new subset of 20 randomly selected species from this set of 61. Again, I really like the effort made to deepen our understanding of broken-wing displays, but I'm not sure I yet believe in the veracity of these data.

Response: Our bootstrapping approach, which we used to obtain confidence bands across the range of predictor variable values for visual display, employs a similar procedure to the second one that the reviewer suggests here. In particular, the bootstrapping procedure runs several analyses on random subsets of the data, leaving data out through the iterations. Therefore, we feel that potential inaccuracies of the survey are not likely to be an issue.

Moreover, while at first sight a published record may seem more valid than a report in a survey, we would argue that this is not necessarily the case. We found the descriptions of the broken wing display in the HBW were often very brief and not backed up by references, thus, they don't strike us as more rigorous than survey results. Ultimately, the description of the broken-wing display in HBW is a trained ornithologist reporting having witnessed the display. Because we distributed our survey to the ornithological community, survey responses are essentially equivalent to descriptions on HBW.

Finally, in an additional effort to address the reviewer's concern that the survey results might be less trust-worthy, we completed additional Google Scholar and Web of Science searches using the terms described in the main text coupled with either the common name or genus species of each species identified in the survey as performing the display. We also re-examined HBW and Birds of the World Online for the same species. The outcome was that we found additional descriptions of the display in the literature or HBW or Birds of the World Online for 37 of the 61 species initially identified by the literature. Many of these were subsequently discovered because the key words mentioned in the main text were not used explicitly to describe the display, but the text clearly describes a display that includes impeded flight due to a feigned injury to the wing. We have now added more information on how the survey was done and which professional ornithologist associations were involved to the supplement and added details of this recursive literature search. We hope these efforts restore this reviewer's trust in the validity of the survey.

Referee: 2

Comments to the Author(s)

This paper seeks to provide a much more comprehensive account of the phylogenetic distribution of an avian anti-predator behavior – the avian broken wing display – than has been available to date.

It's a difficult problem because it's not easy to say definitively that a species lacks this behavior just because there is no reference to it in the literature. Furthermore, there is within species variation, as the paper points out but does not analyze. It's an interesting problem because it appears that the behavior occurs all across the avian tree of life but has blinked on and off many times during evolution. The authors here ask what aspects of life history or ecology are associated with the blinking on and off. They make a set of predictions which I found to be generally reasonable and interesting, and they test those predictions using PGLM.

I found the statistical approach to be curious because, rather than taking the entire set of bird species that are known or not known to do the display, they took each species that is known to do it, and paired it with a close relative that is not known to do it. However, they did not analyze the data as a set of pairs, as might be an effective way to ask what other changes accompany the evolutionary transition to or from using broken wing displays. Rather, they throw all 570 species (285 pairs) into one PGLM. When I started looking through the pairs, I could see perhaps why they didn't do this, as I realized that not all pairs were truly phylogenetically independent. There were very closely related pairs in some cases, such as West Peruvian Dove and White-winged Dove, and very distant ones in other cases, such as Common Nighthawk and Forbes's Plover. Looking at the species that were scored as not using the behavior, I found myself suspecting that many of those species would be discovered in the future as actually using broken wing displays (esp. oystercatcher, plover, and nightjar species); however, I did not see any cases that I could specifically refute. I believe that the literature search methods that were employed were imperfect for this purpose, but I am at least convinced that they were conducted with rigor. One general pattern that was evident was that the member of each species pair that did not use the display was frequently the one that would be expected to be more data deficient on the basis of its distribution and the amount of readily available information about its behavior and life history. The authors were aware of this shortcoming, and they at least partially addressed it by looking at the latitude effect only for high latitudes, a supplementary analysis that I found to be compelling. The latitude result was the most convincing one of this paper in my opinion, because it implies increased nest predation pressure from mammalian predators during the elongated daylight hours of associated with high latitudes.

Response: The reviewer points out many very good points that we have considered as well throughout the course of this project. Also clear from these comments is an implicit understanding that there are no apparent adjustments to the analyses or inclusion of species that would not introduce other issues of concern. We are very grateful to this reviewer for checking our data so thoroughly and for thinking about the drawbacks and advantages of various approaches in such a nuanced way.

It is important to note that we did not use a paired design (as mentioned above, we have clarified our design with more appropriate descriptions). We chose the closest non-displaying species in order to balance the number of displaying and non-displaying species in the analysis. This practice of generating balanced species sets is routinely used in comparative analyses in a variety of sub-fields and represents an adaptation from Felsenstein's (1985. *The American Naturalist* 126: 1-15) phylogenetically independent matched pairs of species approach. For example, in their analysis of song evolution, Tobias et al. (2010. *Evolution* 64: 2820-2839) balanced observations from bamboo-specialist species with their nearest relative in adjacent terra firme forest. In their comparative analyses of endemism and threat status, Martin et al. (2013. *Animal Conservation* 17: 89-96) balanced mammals and bird species with representatives that are held captive in zoos with closely related species that are not represented in zoos. We have added these references to the methods section where we have also clarified our sampling approach (lines 106-109). The alternative approach of using the full set of non-displaying species has two severe shortcomings: (1) it is practically unfeasible to collect the data for all the non-displaying species (more than 9700!). (2) Using a bigger number of non-displaying species increases the probability of including false negatives. Another approach could be to draw a random sample of non-displaying species from across the tree, but this would be even worse in terms of over/underrepresentation of certain clades. Having said this, we don't see how the design of our comparative analysis could be improved. However, we do acknowledge the limitations of our analyses more clearly now in the revised text. Please see also how we have addressed the issue of potential false negatives (our response to Reviewer 3's second point).

A major contribution of this paper was descriptive – what species use the broken wing display and what is their phylogenetic distribution? It follows that an important shortcoming of the paper, in my opinion, was not including visualization of the phylogenetic pattern of presence vs absence of the broken wing display in a way that was more detailed than the family level treatment in figure 2. The phylogenetic pattern of taxon sampling would have been very important to visualize; while I recognize that a tree with 570 tips is too large to fit a page, there are several possible ways around this, and I strongly encourage the authors to come up with one. My preferred option would be to use multiple smaller trees that occupy multiple figure panels or pages. It will be important that the readers can read the taxon names and see the transitions between display and no-display on the phylogeny (a color scheme using red and blue as the authors used in Fig. 2 would be perfect). The visual description is therefore strongly warranted even if it takes a substantial amount of space. I believe that the current figures 1 and 2 are not important to include in the main body of the paper, and I strongly encourage moving them to the supplement in order to make space for a figure that actually displays the taxon sampling scheme and phylogeny that encompass the evolutionary transitions being described and analyzed in this paper.

Response: We have followed the advice to include a more detailed presence/absence figure, but we cannot see how such a figure can be put into the main manuscript and be useful without taking up more than an entire page. Therefore, we added the requested figures to the Supplement. To be clear, we also do not believe that our data can provide the particular insights that this reviewer thinks it could. The transition between presence and absence of the display would not be accurate

in the reduced data set of 572 species. One would need much more data and better knowledge of trait absences to make claims about the precise transitions within particular clades of the avian tree of life. Having said this, we think this would make a very interesting follow-up study.

Specific Comments:

Phylogenetic signal is not addressed in the paper that I could find, but it's clearly relevant and can and should be reported. The models, with the pairwise taxon sampling scheme are likely to underestimate phylogenetic signal, but in the larger context of the avian tree of life, what is the degree of phylogenetic signal, and what is the estimated number of transition events between display and no-display. Figure 2 illustrates the pattern at the family level, but there is no quantification that I could find.

Response: The reviewer brings up a really interesting point about phylogenetic signal of the display and we too would very much like to know what kind of signal exists. Unfortunately, as the reviewer points out, our balanced taxon sampling across clades is likely to muddle any estimate of signal, but of course, estimating the degree of phylogenetic signal in the trait was not the objective of our study. Additionally, this type of approach involves phylogenetic signal of the discrete trait itself rather than accounting for phylogenetic signal in the relationship between predictor and response variables in the models, which, even if lower given our balanced design, is critical to explicitly parameterize in comparative studies. The latter is the phylogenetic signal we estimated on our models. Essentially, we used it to determine whether there is a relationship between predictor variables and the response variable after already accounting for the phylogenetic structure to this relationship. Additionally, although a universal metric for phylogenetic signal that can be comparable across different models and different phylogenies would be nice, the parameter representing phylogenetic signal in the phyloglm function (α) is dependent on branch lengths of a given phylogeny. Because our phylogenies change across subsets in our analyses, α values would not be comparable (nor comparable in any global sense either).

Finally, as mentioned in a previous response, the goal of our paper is not to give an estimated number of the transition events, as this would require more detailed sampling across the entire avian phylogeny. Instead, we aimed to provide an initial, although imperfect, more comprehensive description of the distribution of this behaviour across birds and robust analyses that could identify variables associated with the evolution of the display.

Line 222-223: don't assert sister taxon of turacos, as their phylogenetic placement is deeply problematic and almost certainly wrong here.

Response: Thank you for pointing this out. We have changed the text accordingly.

Prediction table: is the prediction and justification for double-brooding correct? Double brooders have faster life histories (more broods/year), they should also have *more* urgency to protect the present brood. (???)

Response: Our reasoning is that double-brooders have more opportunities to raise offspring and thus should take less risks to defend a single clutch. In contrast, single brooders have only one chance per season to raise offspring and thus should invest more in that one brood. Following this view, the investment in the current reproductive attempt is higher for single-brooded species and they should employ strategies that maximize its success. We are not sure we understand why a faster life history may lead to increased brood protection.

Figure S3 and line 225 and possibly elsewhere: should say “significant association” not “significant influence”; generally be more careful about asserting “influence”.

Response: Thank you for pointing out our overenthusiastic interpretation of correlations. In the revised text we generally use a more cautious and nuanced wording now.

An additional measure to add to the ease of interpretation of this paper would be to have a single figure / table that allows comparison between the predictions (Table 1) and the results.

Response: We have adopted this suggestion by adding a new colour code to Figure S3 that indicates whether the observed effect was in the predicted direction.

There was a survey of expert knowledge conducted, but none of the experts were thanked. I'm surprised at how few references there are here, as if all this information has been effectively purged of its original sourcing. Who else should be acknowledged and cited here? I encourage the authors to consider this deeply before the next stage of publication.

Response: Thank you very much for pointing out our oversight. It wasn't our intention to not give credit where credit is due; it would have been very unfortunate if this information was left out. We now present the details of the survey in the Supplement and thank the experts for their help in the acknowledgements. In addition, we added several citations and we searched the literature again to make sure we didn't overlook any relevant papers.

Referee: 3

Comments to the Author(s)

This manuscript explores the evolution of the broken-wing display, a fascinating distraction and anti-predation behavior that is unique in its complexity and ability to captivate human observers. The authors first identify species that display the trait by using literature searches and surveys, and show that it has likely evolved several times outside the shorebird clades, which are classically associated with the behavior. The manuscript then reviews 16 proposed hypotheses for the evolution of the display, and test predictions using phylogenetically-informed models. The considerable effort to analyze the extent of its evolution is impressive. The manuscript shows that there is still a great deal to be learned about the repeated evolution of this complex trait.

As someone who studies bird behavior myself, I found the study of this trait to be interesting. However, I think the authors could make a much better attempt to persuade a wider audience as to why this trait and its evolution should be of interest. Currently the main justification seems to be that the display is well-known and “textbook”, but this strikes me as a somewhat avian-centric claim. It would help to better use the introduction to place the broken-wing display among other types of deceptive or distractive behaviors. Do non-avian animals use any similar distraction displays during parental care? Why should the audience care about this behavior when it is one of many types of anti-predator strategies used by birds?

Response: We thank the reviewer for this very helpful comment. We now state the broader significance of our study in terms of behavioural adaptation and anti-predator behaviour (see also our response to Reviewer 1’s point 1). By doing so, we address both of the raised issues (introduction lines 69-74). We think the broken-wing display can be used as paradigmatic example to study the evolution of a behavioural trait across an entire class of vertebrates. First, the study of this display allows us to make very clear predictions about the ecological and life history traits that may be associated with it. Second, as Reviewer 1 phrased it, the display is “an incredible example of a “complex behavior” that seems to convergently evolve in a wide range of unrelated avian taxa. This has the potential to shed some interesting light on how behavioral adaptation might work.” In addition, the study of the broken-wing display is highly relevant for our understanding of how deceptive behaviours evolve in the context of predator-prey interactions. Such deception is not specific to birds (many vertebrate species have been described to also perform some similar injury-feigning behaviours), but birds are the one clade in which such a display is most widespread. The broken wing-display is a special case of antipredator behaviour, as it does not function in the protection of the animal exhibiting the display (as is the case in other deceptive behaviours, such as mimicry), but it functions in the protection of the offspring of the displaying animal, a behaviour that is apparently very risky.

Another general concern I have is regarding the high likelihood of false negatives in this study and how that might affect the results. As the authors note in the discussion, many species that exhibit broken-wing displays have likely not been reported. What I believe is still missing, however, is a consideration for how species conspicuousness and rarity will affect the results. This is especially the case if a common or conspicuous species that exhibits the display is “paired”

(line 102) with a "no-display" species that is rare, inconspicuous, or has a small range that would limit observation and increase the probability of a false negative. At the very least, it would be nice to have more discussion on how these issues might bias the results and in which directions. However, one potential way to make these results more convincing would be to include a parameter(s) that is directly related to species visibility to humans. For example, the annual sighting rate of each species on ebird would give a good approximation for how often visible species are (and therefore able to be accurately reported). Including this parameter in the models would allow for the model to account for the issue of conspicuousness and reporting problems.

Response: The reviewer is right, of course. We cannot rule out false negatives, i.e. wrongly assigning displaying species as non-displaying because they are rarely observed and thus no report of the displaying behaviour is available. However, we think that the influence of this possibility is minimal given our separate sensitivity analyses that matched the main findings: (1) analyses using temperate and polar species, which are much better studied and have received considerable attention from the scientific community and, thus, likely contain little to no false negatives. Sub-analyses based on this set yielded basically the same result as the full analysis with the less studied tropical species included (see Supplement). (2) In the light of this comment, we now also conducted a second sensitivity analysis to specifically address the problem of false negatives (provided in the supplement). For this purpose, we analysed only species for which the clutch size or egg measurements are known. The rationale behind this is that to be able to count the eggs, a researcher must have approached the nest, which would have elicited the broken-wing display in one or both parents, provided that the species exhibits this behaviour. It is most unlikely that the display will not be observed when observers approach the nest, as the display behaviour is very conspicuous, and sometimes it is the only breeding behaviour reported for some species. Thus, we are confident that this sub-analysis greatly reduces the potential influence of false negatives. The results of this new sensitivity analysis confirmed our global analysis: the model run with the species with known clutch size indicated the same effects in the same direction as the global model on all species (Table S3). Because the results of both subset analyses match our results from the full analyses for all but one predictor variable (i.e., mobbing), we feel readers can have now more confidence in our findings. We have placed the new subanalyses in the supplement.

Finally, we appreciate the reviewer's suggestion to use eBird or some other measure of visibility to humans. We feel our approach to consider species with known clutch size goes beyond what data from eBird can provide. Many species may be conspicuous to birders but this doesn't necessarily mean they were observed at the nest. In contrast, counting eggs in the nest requires that the observer approached the nest, which would elicit the broken-wing display if the birds exhibit it.

I have also listed line-by-line comments below, most of which are in regards to much-needed clarifications and justifications for various methods.

Lines 90-97: Since the survey added a significant number of species to the dataset, I find myself wanting to know a lot more about the methodology here. Did the authors simply ask participants to specify species that they had seen do a broken-wing display? Were participants told to exclude

species that had already been found in the literature review? What is meant by “environmental context of the observation” (line 91) and how was this information used? How was an observation deemed trustworthy? Were any observations rejected and if so, why? The survey wording itself would be valuable information to include in the supplementary materials. Which internationally-known ornithological organizations were contacted? An analysis of the international distribution of responses would also be informative since biased reporting could play a major role in the results.

Response: We have adopted this suggestion and provide more extensive details on the survey methods in the Supplement.

Line 142: I’m not sure I understand how scrape nests are considered both inconspicuous and also non-protected at the same time.

Response: Nest conspicuousness means how easy it is to spot the nest. Nest protection denotes the protection provided to the eggs by the physical nest structure. Scrape nests are inconspicuous as there is no nest structure, but the eggs are hence directly exposed to the environment, including predators, and thus not protected. We understand that our wording in the methods was not clear enough; now we have rephrased this (lines 140-151) and hope that the ambiguity is resolved.

Line 169-170: Assuming no report of aggressive nest defense reflects an absence of the trait requires us to assume equal observation rates. Is that a safe assumption?

Response: The issue of absence of mobbing or aggressive behaviour is similar to the one of absence of the broken-wing display (see above). We assume that in all species for which the egg data are known, nests have been visited by ornithologists who would have noticed and reported nest defense behaviours, including aggressive behaviour. We are aware that the mobbing probability might change during the course of breeding, but we feel this is the best we can do given the availability of data.

Lines 186-208: This is an intriguing model selection technique that I have not personally seen before. I do not have the expertise to comprehensively judge its competence. Regardless, I believe a great deal of clarification is needed. I do not see any citation to statistical papers or other phylogenetic papers that use a similar technique, so I am unsure how to consider its utility and appropriateness. I understand that the authors were dealing with a variably complete dataset, but this does not clearly justify the method. Other questions I am left with:

- Why did the authors start with only 4 variables? Is there a reason to choose 4?
- Why was only one variable added to make model M1, while it seems 4 variables were added to make model M2? It seems that in most cases variables were added back until 4 variables were reached again, so what’s going on with M2?
- It might help to bold the variables in the “full” models that were $p < 0.1$.

Response: We have clarified these issues in the revised manuscript (lines 197-199, 206-216). In short, we evaluated support for variables in models that differed in sample sizes. This was

necessary because variable completeness reflecting our hypotheses ranged from sample sizes of 569 to 198. We reasoned that it would be unwise to unnecessarily eliminate 65% of samples and only analyze a reduced dataset because larger datasets should provide more precise estimates of the associations between predictor variables and the presence or absence of the broken-wing display. Thus, we started with a model with near complete information for several variables and evaluated relative support at that sample size. Because we found no support for some variables at the largest sample size, such as coloniality, we eliminated this variable from further consideration in models at smaller sample sizes. The rationale is that if we eliminate a variable based on lack of an association with the broken-wing display with a more complete dataset, it does not make sense to include it again in an analysis with a smaller sample size. In contrast, we included variables we found to have strong associations with the broken-wing display in models with smaller sample sizes because it provides the ability to evaluate the relative strength of each variable in the same model and based on the same data.

To explain our approach in more detail, we build the new model at each step based on the completeness of the dataset, and not based on the number of predictor variables. We first ranked all predictor variables in increasing order of sample size and first selected the variables with the largest sample size. These were four variables and therefore all four were included in the model. Then we excluded all uninformative variables using backwards model selection as described in the manuscript. In a second step, we chose the predictor variables with the second highest sample size. We included these in the simplest model of step one, and went through the same simplification procedure to remove all uninformative variables. In a third step, we chose the variables with the third highest sample size, these were four predictor variables with the same sample size. This procedure was repeated for all predictors for which we had a sample size of at least 200 species (198 in supplementary analyses).

-- Line 197-199: Removing an important predictor variable is almost certain to inflate the importance of variables with smaller effect sizes.

Response: We are not convinced that this would necessarily be the case. Carrying insignificant predictors through the analyses would bias the results of subsequent models because this practice unnecessarily increases the model complexity, especially as the sample size is concomitantly reduced. To be consistent with model selection across all sample sizes, we removed any predictor variable (irrespective of whether or not it was associated with the broken-wing display in models with larger sample sizes) as soon as it became uninformative. We have added a justification for the practice of removing predictors once they drop below significance thresholds in line 213.

-- Line 199: It seems to me that collinearity should be checked with the full model before simplification because non-collinearity can lead to incorrect model conclusions.

Response: We completely agree with the reviewer, in fact, this was already part of the procedure. We apologize for not making this point clear. Now we explicitly mention this in lines 215-216.

-- On the same topic, how can the authors be sure collinearity among variables is not an issue? The technique used here never considers a truly full model with all 16 explanatory variables. The results will almost certainly inflate the number of significantly predictive variables because two

correlated variables that are never considered in the same model will both appear to have explanatory power when considered separately.

Response: We are aware of this shortcoming. However, a full model including all variables is not feasible given our variable sample sizes. To overcome this problem, the variables which were highly likely to covary and cause trouble with collinearity were included in the model all together in a single step (e. g. egg mass, size, and brood mass). Collinearity was checked before we removed any variable, and we didn't detect any issues of multicollinearity as assessed with the variance inflation factor. It is true that collinearity was not checked between all possible variable pairs. It is possible that the reviewer's comment is more about not being able to identify which variable is responsible for variation in the occurrence of the broken-wing display when variables in two different models are associated with the broken-wing display. To speak to this concern, we have now added figure S9 to the supplement that illustrates no major issues of correlation among predictors in sub-analysis 2, save for a strong negative correlation between duty cycle and whether the species in multibrooded or not. These two variables were weakly associated with the broken-wing display.

Line 243: Given the very high likelihood that this variable is skewed by biased reporting in these regions, I personally am not very convinced by this sensitivity analysis. Why was 30 chosen? Could the result be spurious? What if a limit of 20, 40, 50, etc. were chosen?

Response: We agree that the reporting of the broken-wing display is likely higher for temperate species than for tropical species. Because of this, we performed the sensitivity analysis on species living in temperate and polar regions. Hence, we would argue that the results are more robust than those of the full analysis. We used 30° as a cut-off to include the well-studied middle latitudes of the temperate zone. The sensitivity analysis, which only contains temperate or polar species, shows the same results as the full analysis, therefore supporting the results of the full analysis.

Supplement:

Line 24: Why use a summarized description rather than the full HBW habitat description? How exactly was a summary made? How is a one-word summary possible?

Response: The habitat descriptions in HBW vary considerably in length: sometimes one single word is used ("rainforest" for example), sometimes, several paragraphs describe the habitat with lots of details (with plant species, lake depth etc.). Therefore, we reduced descriptions to a summary using keywords to avoid having very unbalanced scoring material between species. In the case of long descriptions, we targeted the paragraph describing the breeding habitat (as opposed to wintering habitat) on a medium to large scale, i.e. omitting the very fine scale microhabitat characteristics such as plant species. In this paragraph, we selected the most important keywords of the first three sentences, where the general habitat characteristics were mostly gathered. The following sentences mainly provided more details on the descriptions from the first sentences, therefore, we screened them but retained only new habitats if any. We thank the reviewer for pointing out the lack of clarity in the description of the method used. We have

fixed this issue in the revised Supplement. Since our habitat scores correlate with published measures of habitat cover (see lines 158-160 and supplement), we are confident that our method provides a reliable and creative way to estimate habitat cover from text descriptions.

Appendix B

Associate Editor

Comments to Author:

I commend the authors on successfully addressing the concerns brought up in the review process. As both reviewers recognized, the manuscript has improved tremendously. The addition of the sensitivity analysis, in particular, has helped make a strong case. Overall, the study reads well and provides a more robust exploration of the potential factors associated with the evolution of this interesting behavior.

RESPONSE: Thank you for your kind words

I agree with reviewer 2 that Fig. S3 is a great addition and would be a nice complement to the figures already in the main text. However, in its current form it is really difficult to unpack the information it presents. My concern is that in this figure the reader can easily get a misleading impression of the data. The reviewer provides an excellent description of the main issue of this figure and I strongly recommend re-structuring it to better display the data and support (or lack of support) for the hypotheses. I look forward to seeing the revised version of this figure.

RESPONSE: As suggested by the associate editor and reviewer, we have added a revised version of Fig. S3 and the caption to the main manuscript as a new figure 4. We carefully considered the points raised by reviewer 2 in our revision of the figure and caption. We explored alternative ways of conveying that we found support for the influence of a predictor AND whether the direction of the effect matched the hypotheses presented in Table 1. For instance, in one version we removed the colours and denoted whether the direction of the effect matched the hypothesis by underlining the name of the predictor. This, and, alternatives we considered, however, fails to explicitly point out the parameters that had effects opposite of the hypothesized relationship, which we believe is important because it represents some of the most interesting results. After trying this and other versions, our team ultimately settled on a design similar to what we had before; however, we have heavily revised the figure caption to make it much clearer what the colors and plus/minus signs represent. We hope you find this version satisfactory. Of course, if the editor has particular suggestions of how this figure should be constructed, please let us know. We would be happy to try to incorporate elements that will help the reader understand the information we seek to convey with this figure.

Reviewer(s)' Comments to Author:

Referee: 1

Comments to the Author(s).

The authors make a strong effort to improve the manuscript, and they are successful in many ways. While the same issues still weaken the paper, they are more thoroughly addressed and acknowledged in this version of the manuscript. Hope to see it in print soon!

RESPONSE: Thank you for your kind words.

Referee: 2

Comments to the Author(s).

I was pleased to see the improvements and clarifications to this. I think the phylogeny supplemental figures are helpful, even if not as informative as I had hoped (I'll just note that they look like time-trees but there are no time-scales provided). I would suggest that Fig. S3 (a nice addition!) become Fig. 4 of the main paper. However, there was a weird design decision with Fig. S3 that authors should consider. It took me a few minutes to realize how to interpret it because I had assumed that the +/- symbols would correspond to the direction of the *effect* (which I think is more logical way to read this) when they actually corresponded to the direction of the *hypothesis*. To figure out the direction of the effect, the reader needs to consider both the symbol and the color (a bit convoluted). Anyway, that's a subjective thing, but I'll leave it to the authors to consider changing it. Regarding the issue about our disagreement over a priori expectations around double-brooding and fast life history: my logic related to the tradeoff between parental investment in the current brood versus future ones: double brooding species should tend to have faster life histories in general (lower survival, faster reproduction), and therefore their individual fitness could be more affected by loss of a single brood (whereas a species with a slower life history would survive to breed again the following year); it follows that, for double-brooders, expenditure of energy and risk to avoid loss of a clutch would be more likely to benefit fitness, therefore broken-wing display would be more likely to evolve.

RESPONSE: As mentioned above, we have added a revised version of this figure to the main manuscript as figure 4. Regarding the reviewer's rationale for the parent investment, we think we will have to disagree on this point. We suppose it comes down to the reviewer thinking about lifetime fitness a little differently than we have. We see double brooding as additional opportunities for reproductive attempts, much as longer lifespan would while the reviewer sees

double brooding as a reflection of a faster pace of life with fewer attempts. We don't believe we can reconcile these ways of viewing this issue in this manuscript.

The Dryad data and metadata look excellent and complete.